

# Winter tourism and climate change in the Pyrenees and the French Alps: relevance of snowmaking as a technical adaptation

Pierre Spandre[1,2], Hugues François[1], Deborah Verfaillie[2], Marc Pons[3], Matthieu Vernay[2], Matthieu Lafaysse[2], Emmanuelle George[1], and Samuel Morin[2]

[1]Univ. Grenoble Alpes, Irstea, UR LESSEM, Grenoble, France
[2]Univ. Grenoble Alpes, Université de Toulouse, Météo-France, CNRS, CNRM, Centre d'Études de la Neige, Grenoble, France
[3]Snow and Mountain Research Center of Andorra, Andorra

**Correspondence:** P. Spandre (pierre.spandre@irstea.fr)

**Abstract.** Climate change is increasingly regarded as a threat for winter tourism due to the combined effect of decreasing natural snow amounts and decreasing suitable periods for snowmaking. The present work investigated the snow reliability of 175 ski resorts in France (Alps and Pyrenees), Spain and Andorra under past and future conditions using state-of-the-art snowpack modelling and climate projections. The natural snow reliability (i.e. without snowmaking) elevation showed a

significant spatial variability in the reference period (1986 - 2005) and to be highly impacted by the on-going climate change. The technical reliability (i.e. including snowmaking) is projected to rise by 200 m to 300 m in the Alps and by 400 m to 600 m in the Pyrenees in the near future (2030 - 2050) compared to the reference period for all climate scenarios. While 99% of ski lift infrastructures are reliable in the reference period thanks to snowmaking, a significant fraction (14% to 25%) may be considered "at risk" in the near future. Beyond the mid century, climate projections highly depend on the scenario with

steady conditions compared to the near future (RCP 2.6) or continuous decrease of snow reliability (RCP 8.5). According to the "business as usual" scenario (RCP 8.5), there would no longer be any snow reliable ski resorts based on natural snow conditions in French Alps and Pyrenees (France, Spain and Andorra) at the end of the century (2080 - 2100). Only 24 resorts are projected to remain technically reliable, all being located in the Alps.

## 1 Introduction

The on-going evolution of natural snow conditions related to global climate change (Beniston et al., 2018) is increasingly regarded as a major threat for the winter tourism (Gilaberte-Burdalo et al., 2014; Steiger et al., 2017; Hoegh-Guldberg et al., accepted). This opened a wide discussion on the vulnerability of ski resorts to climate change and the ability of snowmaking to mitigate these effects (Steiger et al., 2017). Initial studies in the early 2000's defined the vulnerability of ski resorts based on a statement referred to as the "100 days" rule and later considered as the reference approach for investigations of climate induced

impacts on the winter tourism (Koenig and Abegg, 1997; Elsasser et al., 2002; Abegg et al., 2007; Steiger, 2010; Pons-Pons et al., 2012; François et al., 2014). This rule states that a ski resort is snow reliable if the snow depth exceeds 30 cm during 100 days or more, featuring the capacity to provide objective information when comparing distinct periods (past and future)





or locations (Koenig and Abegg, 1997; Elsasser et al., 2002; Abegg et al., 2007; Durand et al., 2009b). The snow reliability line is defined as the elevation above which these conditions are met, allowing the assessment of the reliability of a ski resort by comparing its elevation to the snow reliability line (Koenig and Abegg, 1997; Elsasser et al., 2002; Abegg et al., 2007; Gilaberte-Búrdalo et al., 2017).

Most investigations based on the "100 days" rule used single points representations of ski slopes to assess the snow and meteorological conditions of a given ski resort, often using the median elevation of a ski resort defined as the average of summit and base elevations (Abegg et al., 2007; Scott et al., 2003; Steiger, 2010; Dawson and Scott, 2013; Pons et al., 2015; Gilaberte-Búrdalo et al., 2017). Schmidt et al. (2012) and Rixen et al. (2011) used the "highest", "middle" and "lowest" elevations of the study area while Hennessy et al. (2007) mixed various approaches by considering either a single point or three

distinct elevations for each ski resort. Alternatively, Pons-Pons et al. (2012) considered the lowest and highest elevations in which 75% of the ski slopes surface area was concentrated. These remain coarse representations limiting the analysis of the situation of a ski resort to a binary conclusion reliable/unreliable (Steiger et al., 2017). Koenig and Abegg (1997) and Elsasser et al. (2002) in Switzerland and later Abegg et al. (2007) in the rest of the European Alps based their analysis on the natural snow conditions. Abegg et al. (2007) reviewed the existing literature to adress the snow reliability line for regions of Europe

(Austria, Italy, Germany, Slovenia and France) based on distinct methods and reference periods (Laternser and Schneebeli, 2003; Wielke et al., 2004; Matulla et al., 2005). They concluded that 91% of the 666 ski resorts in the European Alps were snow reliable around 2005. Significant spatial variations of the snow reliability line were shown, ranging from 1050 to 1500 m.a.s.l with consequences on local reliability of ski resorts: 69% of ski resorts were snow reliable in Germany and up to 97% in Switzerland and France. Abegg et al. (2007) similarly adressed the impact of climate change on the snow reliability line

and concluded that under a +1°C warming compared to present, only 75% of European Alps ski resorts would remain reliable and respectively 61% and 30% for +2°C and +4°C warming compared to present. These investigations were limited to the analysis of natural snow using average conditions over large regions. Steiger (2010) for example later showed by the analysis of 52 climate stations in Austria over the 1981 - 2001 period that an elevation of 1200 m.a.s.l could not be confirmed as snow reliable for all regions of Tyrol (Austria). Relying on natural snow conditions to assess the snow reliability of ski resorts has

also been questioned, due to the strong role of snow management, in particular grooming and snowmaking (Hanzer et al., 2014; Spandre et al., 2016b; Steiger et al., 2017).

    Recent studies have increasingly taken into account snow grooming and snow making (Scott et al., 2003, 2006; Steiger, 2010; Pons et al., 2015; Steiger et al., 2017). Scott et al. (2003) developed a simple modelling approach accounting for a required snow depth of 50 cm for skiing activities and computed snowmaking requirements based on this target. This method

provided consistent season durations for the 1961 - 1990 reference period in the Southern Ontario region (Canada) which significantly decreased under projected climate conditions despite an increasing need for snowmaking. Scott et al. (2006) later used this modelling approach and a 60 cm snow base depth requirement in the Québec region (Canada). Steiger and Mayer (2008) applied this method in Tyrol (Austria) and concluded that snowmaking could guarantee snow reliability at elevations above 1000 m.a.s.l. for the 1971 - 2000 reference period and would remain a suitable mitigation method until the 2050's with

a significant increase of water and energy requirements (Steiger, 2010). Similar investigations were conducted to assess the



impact of climate change on the ski season duration and the snowmaking requirements so as to compensate the loss over Europe (Damm et al., 2017) or more specifically in regions of Austria (Marke et al., 2014; Hanzer et al., 2014), Germany (Schmidt et al. (2012), Switzerland (Rixen et al., 2011), Andorra (Pons-Pons et al., 2012), Pyrenees (Pons et al., 2015; Gilaberte-Búrdalo et al., 2017), Northeast U.S.A (Dawson and Scott, 2013), New-Zealand (Hendrikx and Hreinsson, 2012), Australia (Hennessy

et al., 2007). Major limitations remain. First, little investigation was undertaken in France, yet a major area for winter tourism (François et al., 2014; Steiger et al., 2017). Second, meteorological and snow input data considered for the analysis were aggregated over large regions (Abegg et al., 2007; Damm et al., 2017) where high spatial variability can be observed (Durand et al., 2009b; François et al., 2014). Third, snow conditions were often simulated using simplified degree day modelling approaches (Dawson and Scott, 2013; Hendrikx and Hreinsson, 2012) and neglected the differences between natural snow and

groomed or machine made snow properties (Pons et al., 2015; Gilaberte-Búrdalo et al., 2017).

The present work aims at producing snow reliability investigations of a wide range of ski resorts in France (Alps and Pyrenees), Spain and Andorra under past and future conditions using state-of-the-art snowpack modelling. We accounted for snow grooming and snowmaking (Spandre et al., 2016b) and used adjusted and downscaled climate projections from the EURO-CORDEX dataset (Verfaillie et al., 2017, 2018) to compute snow reliability elevations with distinct levels of requirements. The

15 mean elevation of residential population in a ski resort (Breiling and Charamza, 1999) and the mean elevation of ski lifts (Falk and Vanat, 2016) were compared to the snow reliability line. We finally provided seven distinct categories for ski resorts based on their natural snow reliability, their degree of dependence on a technical method to achieve reliability (Pons et al., 2015) and whether snowmaking may be a technically efficient method to guarantee snow reliability under present and future climate conditions.

## 2 Method

### 2.1 Ski resorts definition and features

#### 2.1.1 Definition of relevant elevations of ski resorts

All data on the geographical location and technical data on ski resorts were extracted from the "BD Stations" database (François et al., 2014; Spandre et al., 2015). Ski Lifts installation and operation in France are supervised by the STRMTG ("Services

Techniques de Remontées Mécaniques et Transports Guidés"). The STRMTG is a public service in charge of the safety control of French ski lifts providing authorizations for ski lift operations. The STRMTG manages a database (CAIRN: CAtalogue Informatisé des Remontées Mécaniques Nationales) dedicated to ski lifts which includes technical characteristics of each ski lift such as the ski lift power. The ski lift power is an indicator of the size of a ski lift, defined as the product of the elevation difference between the bottom and the top of a ski lift (in km) and its flow of persons per hour (pers h$^{-1}$), expressed in pers

30 km h$^{-1}$. Ski lifts infrastructures in France have a total ski lift power of 977 000 pers km h$^{-1}$, 94% of which are included in the present study (Appendix B). These data are completed with geographical information from the database BDTOPO (25 m of resolution) developed by the French Geographical Institute (IGN, "Institut Géographique National"). Ski resorts operated by a





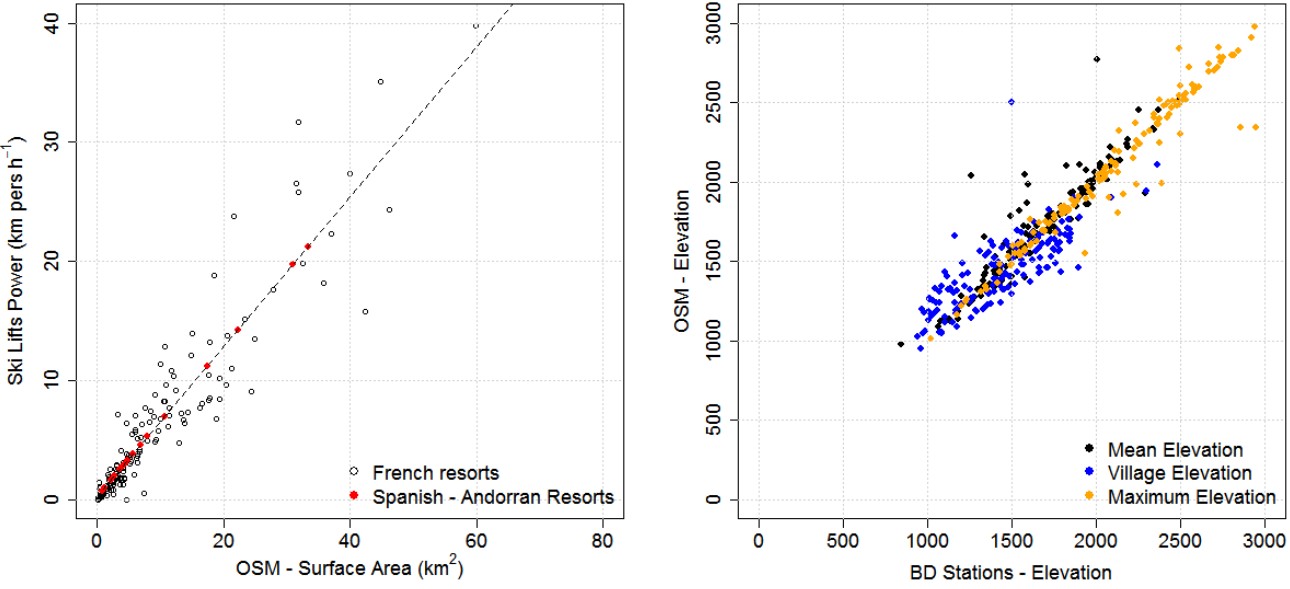

**Figure 1.** Evaluation of OpenStreetMap (OSM) data on French ski resorts and estimation of Spanish and Andorran ski resorts features based on linear models for the ski lift power and elevations of the ski resort (min, mean, max).

single company are aggregated in the present study (François et al., 2014) e.g. Stations du Mercantour (Isola 2000, Auron, La Colmiane). The following elevations were used to be compared with the snow reliability line:

- The ski lifts mean elevation is defined as the average of top and bottom elevations of each ski lift weighted by its ski lift power, being simply referred to as the mean elevation of the ski resort (François et al., 2014; Falk and Vanat, 2016).

- The village elevation of a ski resort is defined as the average elevation of all urban areas within 300 m of the bottom of a ski lift, weighted by their surface area (Breiling and Charamza, 1999).

Based on the OpenStreetMap project (http://www.opensnowmap.org/) we estimated the main features for the Spanish and Andorran ski resorts (village elevation, mean ski lift elevation and ski lift power). We extracted the ski slopes from the open source database for all ski resorts in France, Spain and Andorra and estimated these parameters from the area resulting of a
10 positive/negative buffer and intersection to define a spatial representation for every ski resort. Linear models were fitted based on French ski resorts and applied to Spanish and Andorran ski resorts to provide theses features (Figure 1).

- The linear model of the ski lift power versus the OpenStreetMap surface area (Figure 1 left) had a correlation coefficient $R^2 = 0.87$ (p-value $< 10^{-15}$), proving relevant to estimate the ski lift power based on the OpenStreetMap surface area.

- Elevations derived from the OpenStreetMap spatial representation also proved significantly correlated to data from the
15 BD Stations (Figure 1 right):

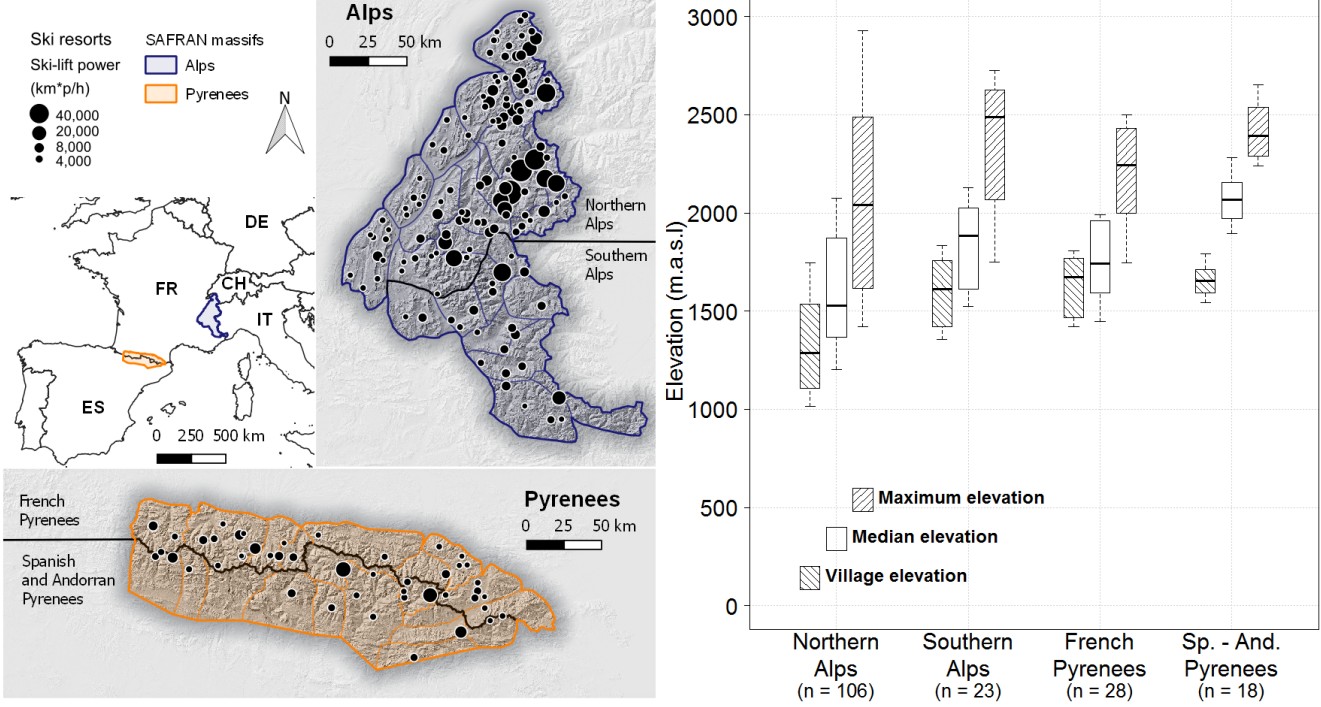

**Figure 2.** (left) The 175 ski resorts covered by the present study and the 44 massifs from the SAFRAN meteorological system and (right) Distribution of ski resorts elevations depending on their location: Northern Alps, Southern Alps, French Pyrenees and Spanish and Andorran Pyrenees ("Sp. - And. Pyrenees"). See Appendix B

- All elevations together: RMSD = 149 m, mean difference = 15 m.
  Linear model of slope 0.97 ($R^2$ = 0.91, p-value $< 10^{-15}$).

- Mean elevation: RMSD = 154 m, mean difference = 51 m.
  Linear model of slope 0.82 ($R^2$ = 0.83, p-value $< 10^{-15}$).

- The village elevation proved significantly correlated to the mean elevation derived from OpenStreetMap spatial representations (slope 0.64, intercept 326 m, $R^2$ = 0.62, p-value $< 10^{-15}$). The linear model was applied to estimate the village elevation from the OpenStreetMap mean elevation and compared to the BD Stations data: RMSD = 179 m, mean difference $< 10^{-12}$ (Figure 1 right).

### 2.1.2 Study area

10  A sample of 175 ski resorts in the French Alps (n = 129), the French Pyrenees (n = 28), the Spanish Pyrenees (n = 14) and Andorra (n = 4) were included in the present study (Figure 2, Appendix B). The French ski resorts included in this study (n = 157) represent 94% of the national ski lifts infrastructures.



## 2.2 Definition and computation of the Snow Reliability Line

### 2.2.1 Snowpack modelling

The "Crocus Resort" version of the multilayer snowpack model SURFEX/ISBA - Crocus was used in the present study. SUR-FEX/ISBA - Crocus is an open source model available online at https://opensource.umr-cnrm.fr/ (Brun et al., 1992; Vionnet
et al., 2012). Crocus Resort allows taking into account the grooming and snowmaking effect on snow properties so as to provide simulations of snow conditions on ski slopes (Spandre et al., 2016b). The impacts of grooming are simulated and machine made snow can be added to the snowpack specifying the precipitation rate ($1.2 \ 10^{-3}$ kg m$^{-2}$ s$^{-1}$, Spandre et al. (2016a)) and conditions for triggering the production (wet-bulb temperature threshold -2°C, target quantity or target snow depth). The production of snow was based on the following rules, dividing the winter season into distinct periods (Steiger, 2010; Hanzer et al., 2014; Spandre et al., 2016a):

- Between November 1 and December 15, a 30 cm deep "base layer" (150 kg m$^{-2}$) is produced, weather conditions permitting, regardless of natural snowfalls during the period.

- Between December 15 and February 28, snow is produced, if meteorologically possible, so as to maintain a total snow depth of 60 cm.

- After March 1, no more snow is produced.

### 2.2.2 Climate forcing data

The meteorological system SAFRAN (Durand et al., 1993) provides meteorological data (temperature, precipitations, etc.) for mountain areas of an approximate 1000 km$^2$ surface referred to as "massif", covering French Alps and Pyrenees, including Spanish and Andorran Pyrenees (Figure 2). Within each massif, the meteorological conditions are supposed to be homogeneous and to depend only on the altitude (300 m altitudinal step) with a time resolution of 1 h. SAFRAN forcing data are available for the 1958 - 2015 period (Durand et al., 2009a; Maris et al., 2009; Durand et al., 2012). Computations of snow conditions over the reference period using SAFRAN forcing data are further referred to as "SAFRAN" and can be considered as the reference observational dataset.

This study uses the EURO-CORDEX dataset (Jacob et al., 2014; Kotlarski et al., 2014) for climate projections consisting of six regional climate models (RCMs) forced by five different global climate models (GCMs) from the CMIP5 ensemble (Taylor et al., 2012) over Europe, for the historical, RCP 2.6 , RCP 4.5 and RCP 8.5 scenarios (Moss et al., 2010). All EURO-CORDEX data were adjusted using the ADAMONT method (Verfaillie et al., 2017) to the mountain areas defined for the SAFRAN system. Historical runs generally cover the period 1950 - 2005 and RCPs cover the period 2006 - 2100 (Table 1). Continuous hourly resolution meteorological time series derived from RCM output by the ADAMONT statistical adjustment method are then used as input of the SURFEX/ISBA-Crocus snowpack model (Verfaillie et al., 2017, 2018).



| RCM (institute)/GCM | Period | CNRM-CM5 | EC-EARTH | HadGEM2-ES | MPI-ESM-LR | IPSL-CM5A-MR |
|---|---|---|---|---|---|---|
| CCLM 4.8.17 (CLMcom) | HIST | 1950-2005 | 1950-2005 | 1981-2005 | 1950-2005 | |
| | RCPs | 2006-2100 | 2006-2100 | 2006-2099 | 2006-2100 | |
| ALADIN 53 (CNRM) | HIST | 1950-2005 | | | | |
| | RCPs | **2006-2100** | | | | |
| WRF 3.3.1F (IPSL-INERIS) | HIST | | | | | 1951-2005 |
| | RCPs | | | | | 2006-2100 |
| RACMO 2.2E (KNMI) | HIST | | | 1981-2005 | | |
| | RCPs | | | **2006-2099** | | |
| REMO 2009 (MPI-CSC) | HIST | | | | 1950-2005 | |
| | RCPs | | | | 2006-2100 | |
| RCA 4 (SMHI) | HIST | 1970-2005 | 1970-2005 | 1981-2005 | 1970-2005 | 1970-2005 |
| | RCPs | 2006-2100 | **2006-2100** | 2006-2099 | 2006-2100 | 2006-2100 |

**Table 1.** EURO-CORDEX GCM-RCM combinations used in this study (rows: RCMs; columns: GCMs), with the time period available for the HIST and RCP 4.5 and 8.5 scenarios (RCPs). Model combinations additionally using RCP 2.6 are displayed in bold. Contributing institutes are indicated inside parentheses - CLMcom: Climate Limited-area Modelling community with contributions by BTU, DWD, ETHZ, UCD,WEGC; CNRM: Météo France; IPSL-INERIS: Institut Pierre Simon Laplace, CNRS, France - Laboratoire des Sciences du Climat et de l'Environnement, IPSL, CEA/CNRS/UVSQ - Institut National de l'Environnement Industriel et des Risques, Verneuil en Halatte, France; KNMI: Kingdom of Netherlands Meteorological Institute, Ministry of Infrastructure and the Environment; MPI-CSC: Max Planck Institute for Meteorology, Climate Service Center, Hamburg, Germany; SMHI: Swedish Meteorological and Hydrological Institute, Rossby Centre, Norrkoping Sweden

### 2.2.3 Snow indicators

The snow reliability line was computed from the simulated snow conditions for the reanalysis and all GCM/RCM pairs and scenarios. The snow reliability line was based on the "100 days rule" and defined for a given season as the elevation above which a minimum quantity of 100 kg m$^{-2}$ of snow was simulated during at least 100 days between December 15 and April 15 (Scott

5 et al., 2003; Steiger, 2010; Marke et al., 2014; Pons et al., 2015). The use of snow mass instead of snow depth (Marke et al., 2014) appeared more relevant in considering the differences between the natural snow properties and the machine made snow or even groomed snow (Spandre et al., 2016b). Based on the season length computed for SAFRAN massifs elevations (300 m step), a linear interpolation was used to compute the snow reliability line meeting the 100 days threshold. In case the season length at the minimum (respectively maximum) elevation proved higher (respectively lower) than 100 days, the snow reliability

10 line was set to half the altitudinal step (150 m) below (respectively above) the minimum (respectively maximum) elevation for a



given massif. We further computed for each massif the snow reliability line by considering distinct periods, climate scenarios, snow requirements and snow management, providing 48 distinct values of the snow reliability elevation resulting from the combination of these parameters (Tables A1, A2, A3 in appendix). Eight periods and scenarios configurations are based on the reference period (1986 - 2005) using the SAFRAN Reanalysis and available GCM/RCM pairs (HIST), the near future (2030

5 - 2050) and the end of the century (2080 - 2100), using climate scenarios RCP 2.6, RCP 4.5 and RCP 8.5 for all available GCM/RCM pairs (Table 1). Three distinct levels of snow reliability requirements were defined as the elevation where the season length reached 100 days one season out of two (50% percentile of annual values), seven seasons out of ten (70% percentile of annual values) and nine seasons out of ten (90% percentile of annual values). Last, we considered the groomed snow conditions (no snowmaking) and including snowmaking (two configurations).

## 2.3   Definition of snow reliability categories

Seven snow reliability categories have been designed with respect to the natural snow reliability and the relevance of snowmaking as an efficient mitigation method to tackle snow variability and scarcity, in line with previous investigations (Pons et al., 2015; Steiger and Mayer, 2008). Following Steiger and Mayer (2008) we considered a strict threshold of nine winters out of ten for snowmaking reliability (90% percentile of annual values), considering that snowmaking facilities are an investment for

the operability of ski lifts and should therefore target a high level of reliability. The following categories were defined to characterize the snow reliability of ski resorts, depending on the relationship between village elevation and mean ski lift elevation, and the reliability lines with and without snowmaking. Categories are ordered by decreasing levels of natural and managed snow reliability. For each ski resort, its category corresponds to the first one for which the criterion is fulfilled, starting from Category 1 until Category 7. A ski resort fullfilling the condition of category N-1 also fullfills the condition of category N. Ski

resorts in category N fulfill the condition of category N but not the condition of category N-1.

- Category 1: Village elevation above the 90% groomed snow reliability line

- Category 2: Village elevation above the 70% groomed snow reliability line and village elevation above the 90% snowmaking reliability line

- Category 3: Ski lifts mean elevation above the 70% groomed snow reliability line and village elevation above the 90%
snowmaking reliability line

- Category 4: Ski lifts mean elevation above the 50% groomed snow reliability line and village elevation above the 90% snowmaking reliability line

- Category 5: Village elevation above the 90% snowmaking reliability line

- Category 6: Ski lifts mean elevation above the 90% snowmaking reliability line

- Category 7: Ski lifts mean elevation below the 90% snowmaking reliability line

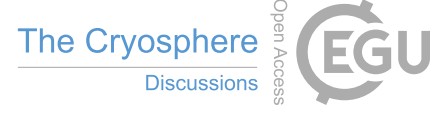

Categories 1, 2 and 3 illustrate ski resorts where natural snow conditions are reliable (Abegg et al., 2007; Scott et al., 2003; Pons et al., 2015). Snowmaking is limited to the lowest elevations and for a minority of seasons. Categories 4 and 5 illustrate ski resorts where natural snow conditions may not be considered as reliable but snowmaking can guarantee the reliability in all elevations of the resort. In these two categories, snowmaking is useful and efficient in mitigating natural snow scarcity at all elevations of the resort providing "technical" reliability (Pons et al., 2015). Categories 6 and 7 illustrate ski resorts where natural snow conditions may not be considered as reliable and snowmaking is no longer efficient in mitigating natural snow scarcity at the lowest elevations of the resort.

## 3 Results

### 3.1 Snow conditions and snow reliability line

#### 3.1.1 Spatial variability

A significant spatial variability of the snow reliability line can be observed for the reference period (1986 - 2005), the median elevation of the 70% groomed snow reliability ranging between 1750 m.a.s.l in the Northern Alps, 2000 m.a.s.l in the French Pyrenees, 2250 m.a.s.l in the Southern Alps and up to 2300 m.a.s.l in the Spanish and Andorran Pyrenees (HIST, Figure 3). Although a bias can be observed, the spatial variability is consistent between climate models computations over the reference period (HIST) and the reference dataset (SAFRAN). The 90% snow reliability using snowmaking is significantly lower than the 70% groomed snow reliability line (Figure 3). The gain thanks to snowmaking on the median elevation ranges between 700 m in the French Pyrenees, 900 m in the Spanish and Andorran Pyrenees, 1000 m in the Northern Alps and up to 1200 m in the Southern Alps. This results in a technical reliability line significantly lower in the Southern Alps compared to the Pyrenees despite poorer natural snow conditions (Figure 3). Although the improvement of snow conditions thanks to snowmaking is lower in the Pyrenees compared to the Alps, the annual snowmaking requirements are higher with 400 to 550 kg m$^{-2}$ machine made snow produced at the snow reliability line in the Northern and Southern Alps (10% - 90% percentiles of annual values) and 400 to 700 kg m$^{-2}$ in the French, Spanish and Andorran Pyrenees (HIST). Such production is equivalent to 80 cm to 1.1 m of snow in the Alps and 80 cm to 1.4 m of snow in the Pyrenees at the snow reliability line (machine made snow density of 500 kg m$^{-3}$).

#### 3.1.2 Evolution

Natural snow conditions are significantly impacted by climate change in the near future (2030 - 2050) with similar evolutions between climate scenarios (Figure 3). The median 70% groomed snow reliability line is projected to range between:

- 1850 m.a.s.l and 2000 m.a.s.l in the Northern Alps
  (100 to 250 m above the reference period)





**Figure 3.** Spatial variability between massifs and evolution for the reference period, the near future (2030 - 2050) and the end of the century (2080 - 2100) of the snow reliability line based on RCP 2.6, RCP 4.5 and RCP 8.5 for the main areas covered in the present study (Northern and Southern Alps, French And Spanish - Andorran Pyrenees)

- − 2500 m.a.s.l and 2650 m.a.s.l in the Southern Alps
  (200 to 400 m above the reference period)

- − 2250 m.a.s.l and 2300 m.a.s.l in the French Pyrenees
  (300 to 350 m above the reference period)

5  − 2550 m.a.s.l and 2650 m.a.s.l in the Spanish and Andorran Pyrenees
  (300 to 350 m above the reference period)

Due to the combined effect of decreasing natural snow conditions and decreasing suitable conditions for snowmaking, the 90% snow reliability line using snowmaking is projected to rise by 200 m to 300 m in the Northern Alps, 300 m in the Southern Alps and up to 400 m to 600 m in the Pyrenees compared to the reference period. In the near future the median elevation of the





technical reliability is projected to range between 950 m.a.s.l to 1050 m.a.s.l in the Northern Alps, 1350 m.a.s.l in the Southern Alps, 1700 m.a.s.l to 1850 m.a.s.l in the French Pyrenees, 1750 m.a.s.l to 1900 m.a.s.l in the Spanish and Andorran Pyrenees. The production of machine made snow at the snow reliability line is projected to remain steady or to decrease in the Pyrenees, up to 15% compared to the reference period on the contrary to the Alps where snowmaking is projected to increase for all

scenarios up to 15%, either in the Northern or Southern Alps. This highlights the higher suitability of climate conditions for snowmaking in the Alps compared to the Pyrenees and increases the gap in the elevation of the technical reliability between these areas (Figure 3).

The impact of climate change on the natural snow conditions beyond the mid century is projected to be highly dependent on the climate scenario. Conditions at the end of the century (2080 - 2100) are projected to remain similar to the near future for

RCP 2.6, the median 70% groomed snow reliability line ranging between 200 m to 300 m above the elevation for the reference period. According to the RCP 8.5, this elevation at the end of the century would be 850 m higher than the value for the reference period in the Northern and Southern Alps, 900 m in the Spanish and Andorran Pyrenees, up to 1050 m in the French Pyrenees. The technical reliability elevation may suffer from the decreasing suitable periods for snowmaking, the median elevation at the end of the century ranging 200 m (Northern Alps) to 450 m (French Pyrenees) higher than the value for the reference period

for the RCP 2.6 and up to 1100 m (Northern and Southern Alps) to 1450 m (French Pyrenees) higher for the RCP 8.5. The median elevation of the technical reliability for the RCP 8.5 would range at the end of the century between 1850 m.a.s.l in the Northern Alps, 2150 m.a.s.l in the Southern Alps and 2700 m.a.s.l in the French, Spanish and Andorran Pyrenees (Figure 3). In the Pyrenees, the production of machine made snow is projected to decrease by 15% to 35% in the French Pyrenees and 10% to 20% in the Spanish and Andorran Pyrenees (10% - 90% percentiles) compared to the reference period due to the lack of

suitable conditions. In the Alps, snowmaking is projected to remain relatively steady at the snow reliability elevation compared to the near future with higher requirements compared to the reference period up to 10%.

## 3.2   Reliability of ski resorts

Figures 3 and 4 and Table 2 show a bias between the SAFRAN reference dataset and results derived from climate models (HIST) for the reference period. We therefore focus our analysis on the comparison of snow conditions computed by climate

models for the reference and future periods. Based on climate models, ski lifts infrastructures were reliable during the reference period (1986 - 2005), either with natural snow conditions (50% in categories 1, 2 and 3 all together, Table 2, HIST) or technically (49% in categories 4 and 5 all together). Natural snow conditions in larger ski resorts were more reliable than in the smaller ones with 44 resorts representing 50% of the ski lift power being natural snow reliable and 129 ski resorts also representing 49% of the ski lift power being only technically reliable (Table 2). Categories 6 and 7 include resorts where 90%

technical reliability can not be achieved at the elevation of the village (category 6) or at the ski lifts mean elevation (category 7). These categories represent the situation of a marginal fraction of ski resorts in the reference period: less than 1% unreliable facilities (2 resorts in these categories) and might therefore be considered "at risk" in terms of snow conditions. Figures 4 and 5 also illustrate a significant geographical pattern with most natural snow reliable ski resorts being located in the Northern Alps and central Pyrenees. This can be related to the lower elevation of the snow reliability line in the Northern Alps compared to



| Category | Reference (1986 - 2005) | | Near Future (2030 - 2050) | | | End of the Century (2080 - 2100) | | |
|---|---|---|---|---|---|---|---|---|
| | SAFRAN | HIST | RCP 2.6 | RCP 4.5 | RCP 8.5 | RCP 2.6 | RCP 4.5 | RCP 8.5 |
| 1 | 21 (n = 11) | 2 (n = 2) | 0 | 2 (n = 2) | 0 | 0 | 0 | 0 |
| 2 | 7 (n = 15) | 13 (n = 7) | 2 (n = 2) | 8 (n = 3) | 5 (n = 3) | 2 (n = 2) | 2 (n = 2) | 0 |
| 3 | 44 (n = 53) | 35 (n = 35) | 19 (n = 12) | 22 (n = 19) | 21 (n = 14) | 25 (n = 16) | 7 (n = 4) | 0 |
| 4 | 16 (n = 42) | 27 (n = 51) | 29 (n = 25) | 23 (n = 24) | 19 (n = 19) | 20 (n = 23) | 27 (n = 20) | 4 (n = 2) |
| 5 | 13 (n = 50) | 22 (n = 78) | 35 (n = 91) | 31 (n = 81) | 30 (n = 64) | 41 (n = 90) | 33 (n = 63) | 24 (n = 22) |
| 6 | 0 (n = 4) | 1 (n = 2) | 13 (n = 31) | 12 (n = 29) | 18 (n = 39) | 10 (n = 28) | 16 (n = 35) | 21 (n = 21) |
| 7 | 0 | 0 | 2 (n = 14) | 2 (n = 17) | 7 (n = 36) | 2 (n = 16) | 14 (n = 51) | 51 (n = 130) |

**Table 2.** Distribution of the total ski lift power (%) within reliability categories for distinct periods and scenarios (Figure 4) with the number of resorts included (n).

the Southern Alps or the Pyrenees (Figure 3, Appendix A1) and the higher elevation of larger ski resorts, most of them being located in the Northern Alps and central Pyrenees (Figures 2 and 5). The variability is particularly high between Northern Alps (a majority of ski resorts natural snow reliable: 67% of ski lift power) and the Southern Alps (89% being only technically reliable) highlighting a higher dependence of Southern Alps ski resorts to snowmaking in the reference period (only 12% of ski lift power natural-snow reliable). The situation of the Pyrenees ski resorts lies in between (Figure 5).

In the near future (2030 - 2050) and depending on the RCP, only 14 to 24 ski resorts (21 to 32% of ski lift power) are projected to remain snow reliable based on natural conditions, all being located in the Northern Alps except one in central Pyrenees (Table 2). An additional 83 to 116 resorts (representing 49 to 64% of ski lift power) are projected to remain technically reliable thanks to snowmaking. Overall, a majority of ski resorts would remain reliable, either technically or under natural snow conditions (75 to 86% of ski lift power). A significant fraction of 45 to 75 ski resorts (14 to 25% of ski lift power) would however turn either into category 6 (12 to 18% of ski lift power) or even in category 7 (2 to 7% of ski lift power) where 90% technical reliability can not be achieved at the elevation of the village (category 6) or at the ski lifts mean elevation (category 7). The geographical pattern is projected to remain in the near future. Even though there would not be any natural snow reliable ski







**Figure 4.** Distribution of the total ski lift power (%) within reliability categories for distinct periods and scenarios (Table 2)

resort in the Southern Alps, snow conditions are projected to remain technically reliable for most resorts (100% to 89% of technically reliable ski lift power), displaying a consistent distribution between reliability categories compared to the reference



period (Figure 4). On the contrary, the projected impact on the Pyrenees ski resorts is significant, particularly in the French Pyrenees. There would remain a single resort being natural snow reliable but more important is the fraction of resorts turning into category 6 (45 to 58% of ski lift power in the French Pyrenees and 32 to 59% in the Spanish and Andorran Pyrenees) or even category 7 (12 to 42% of ski lift power in the French Pyrenees and 7 to 20% in the Spanish and Andorran Pyrenees).

Beyond the near future, the evolution of snow conditions strongly depends on the climate scenario, due to both the evolution of natural snow conditions and on the availability of suitable periods for snowmaking (Figure 3). According to the scenario RCP 2.6, snow reliability may remain similar or even improve at the end of the century (2080 - 2100) compared to the near future (2030 - 2050). Figures 3 and 4 and Table 2 illustrate the significant impact of climate change on the snow conditions and ski resorts reliability for the RCP 8.5 compared to the two other scenarios. There would not remain any ski resort with

reliable natural snow conditions based on the RCP 8.5 with only 24 ski resorts (28% of ski lift power) benefiting from technical reliability (Table 2), all of them being located in the Alps. Figure 4 illustrate a strong geographical pattern within the Alps with higher snow reliability in Eastern central Alps compared to external and Southern massifs. Technically reliable ski resorts are located in Vanoise (n = 7), Haute-Tarentaise (n = 5), Maurienne (n = 5) and Haute-Maurienne (n = 3) in the Northern Alps, and Thabor (n = 1), Pelvoux (n = 1), Queyras (n = 1) and Champsaur (n = 1) in the Southern Alps.

## 15    4   Discussion

A number of limitations remain in our approach and should be carefully considered in the interpretation of our results. Concerning the modelling of the snowpack evolution under past and future climate conditions, meteorological forcing data are aggregated at the scale of a massif (an approximate 1000 km$^2$ surface area) and by elevation steps of 300 m which is a significant improvement compared to previous investigations (Abegg et al., 2007; Damm et al., 2017) although local effects are still

neglected. The snow melting rate is probably underestimated in the model leading to optimistic results (Spandre et al., 2016b). The main reason for this is the one dimensional assumption in the snowpack model thereby neglecting the snow/ground partitioning, particularly when the natural snow melts out and leaves the ski slope as an isolated snow patch in grass or rock fields (Mott et al., 2015). This situation is likely to be more frequent under future climate conditions resulting in increasingly optimistic results compared to the reference period. Additionally, all results computed based on the observational reference

dataset and climate models exhibit differences in the reference period (Figures 3 and 4 and Table 2). Discrepancies may be due to potential biases of the multivariate distribution of the meteorological variables produced by the adjustment and downscaling method (Verfaillie et al., 2017). This could result in potential nonlinear effects due to multiple dependencies especially on temperature, relative humidity, precipitation and wind-speed.

       Beyond the modelling of snow conditions the main limitations pertain to the snow reliability line approach. Single points

representations are considered on flat field i.e. neglecting the aspect and slope angles of a given ski area which is of high importance in the seasonal evolution of the snowpack and might highly differ from a resort to another. These representations also neglect that all slopes are not covered by snowmaking facilities hampering any detailed investigation of the evolution of water requirements (results are limited to values per unit surface area). Modelling chains including spatial representations of





**Figure 5.** Fraction of ski lift power (%) for a given category (Section 2.3). Categories 1, 2 and 3 illustrate ski resorts where natural snow conditions are reliable. Categories 4 and 5 illustrate ski resorts where snow conditions are technically reliable. Categories 6 and 7 illustrate ski resorts where snowmaking is no longer efficient in mitigating natural snow scarcity at the lowest elevations of the resort.

ski resorts may overcome such weaknesses of the snow reliability line approach (Spandre et al., Under Review). Additionally, even though snowmaking may appear as an efficient method to technically mitigate the impacts of natural snow scarcity, the attractiveness of a given resort may be damaged either due to the lack of snow where not equipped with facilities or even due the lack of natural snow (landscapes, winter spirit). We also provided information beyond a binary assessment reliable/unreliable 5 by creating reliability categories, although economic implications should be specifically investigated with a more detailed approach. The relative economic importance of specific periods (Christmas and Winter school holidays) is also neglected in this approach, similarly to previous uses of the snow reliability line.





## 5 Conclusion

State-of-the-art snowpack modelling and climate projections were used in the present investigation to provide a snow reliability assessment of a large sample of 175 ski resorts in French Alps and Pyrenees (France, Andorra and Spain) under past and future climate conditions. A significant spatial variability was highlighted. The Northern Alps showed the best natural snow conditions

either for the reference period and under future climate conditions. Snowmaking appears as an efficient method to improve the snow reliability with 99% of ski lift facilities technically reliable for the reference period (1986 - 2005). This is particularly true in the Southern Alps where snowmaking leads to lower elevation of the technical reliability compared to the Pyrenees while natural snow reliability line is higher. This situation is projected to remain in future climate conditions and snow reliability elevation may significantly rise due to the decrease of natural snow conditions and of the suitable conditions for snowmaking.

The projected deviation between climate scenarios impacts is very low in the near future (2030 - 2050). Depending on the RCP, 21 to 32% of ski lift infrastructures would remain reliable based on natural snow conditions while another 14 to 25% might be considered "at risk" i.e. for which technical reliability can not be achieved. Significant snowmaking requirements are projected to be necessary at the snow reliability line ranging between 400 and 700 kg m$^{-2}$ i.e. an equivalent 80 cm to 140 cm machine made snow production. Deviations between climate scenarios only appear after the mid century with limited

evolutions compared to the near future (RCP 2.6) or continuous decrease of the snow reliability (RCP 8.5). At the end of the century and for the RCP 8.5, there would not remain any reliable resort based on natural snow conditions and only 24 resorts (28% of ski lift facilities) benefiting from technical reliability, all being located in the Alps. The past and future snow reliability of ski resorts in the French Alps and Pyrenees is highly variable, the present investigation showing the relevance of considering local situations rather than general conclusions. We believe that our results might be a substantial material for discussions of

the relevance of snowmaking as a technical adaptation and the decision making for investments in these facilities. Management implications and economic issues might also be derived from this approach which should be extended to mid elevation areas in France (Jura, Vosges, Massif Central). This also bears the potential for wider extension including at the European scale taking advantage of the fact that the method does not require complex data to characterize ski resorts (village and mean ski lift elevation) and could be applied to ongoing simulations of natural and managed snow at the European scale (Morin et al.,

2018).

*Author contributions.* SM and EG designed the research; PS developed the model, carried out the experiments and produced the data and most figures with support from co-authors; HF contributed to produce the data and mapping figures; DV produced the adjusted climate projections; ML contributed to the development of the model and production of climate forcing data; MV contributed to the production of the reanalysis forcing data; all authors contributed to the analysis and interpretation of the results; PS wrote the paper, based on input and

30 feedback from all co-authors.

*Competing interests.* The authors declare that there is no conflict of interest regarding the publication of this article.



*Acknowledgements.* This study was funded by Région Rhône-Alpes (PhD grant of Pierre Spandre), and benefited from funding from the French Ministry for Ecology (MTES) to the ADAMONT project through GICC and ONERC, from the Interreg project POCTEFA/Clim'Py, from the IDEX Univ. Grenoble Alpes Cross Disciplinary Project "Trajectories", and from the Conseil Départemental de l'Isère - Isère Tourisme. The CGET - Comité de Massif Alpes funded the creation of the BD Stations database. CNRM/CEN and Irstea are part of LabEX
5   OSUG@2020 (ANR10 LABX56).





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




## Appendix A: Snow reliability elevation

### A1  Reference Period (1986 - 2005)

Table A1 Snow reliability elevation for the reference period (1986 - 2005) for the 42 massifs distributed over the Northern Alps, Southern Alps, French Pyrenees and Spanish and Andorran Pyrenees, computed by the reference dataset (SAFRAN) and the climate models (HIST) for three distinct reliability requirements (50%, 70% and 90%).

| | Groomed Snow | | | | | | Including Snowmaking | | | | | |
| | SAFRAN | | | HIST | | | SAFRAN | | | HIST | | |
| | (Quantiles) | | | (Quantiles) | | | (Quantiles) | | | (Quantiles) | | |
| Massif | 50% | 70% | 90% | 50% | 70% | 90% | 50% | 70% | 90% | 50% | 70% | 90% |
|---|---|---|---|---|---|---|---|---|---|---|---|---|
| **Northern Alps** | | | | | | | | | | | | |
| Chablais | 1240 | 1410 | 1630 | 1350 | 1580 | 1940 | 450 | 670 | 930 | 450 | 630 | 780 |
| Aravis | 1220 | 1370 | 1620 | 1310 | 1540 | 1910 | 750 | 750 | 750 | 750 | 750 | 750 |
| Mont-Blanc | 1160 | 1390 | 1580 | 1350 | 1580 | 1930 | 1050 | 1050 | 1050 | 1050 | 1050 | 1050 |
| Bauges | 1190 | 1440 | 1670 | 1340 | 1590 | 1970 | 450 | 450 | 650 | 450 | 450 | 730 |
| Beaufortain | 1270 | 1430 | 1660 | 1350 | 1620 | 2100 | 750 | 750 | 750 | 750 | 750 | 750 |
| Haute-Tarentaise | 1450 | 1560 | 1720 | 1470 | 1800 | 2280 | 750 | 750 | 750 | 750 | 750 | 750 |
| Chartreuse | 1310 | 1490 | 1740 | 1420 | 1830 | 2070 | 450 | 680 | 770 | 450 | 650 | 860 |
| Belledonne | 1380 | 1510 | 1650 | 1420 | 1650 | 1960 | 450 | 650 | 770 | 450 | 450 | 750 |
| Maurienne | 1450 | 1550 | 1740 | 1420 | 1740 | 2160 | 450 | 620 | 780 | 450 | 450 | 690 |
| Vanoise | 1490 | 1690 | 1780 | 1460 | 1830 | 2230 | 750 | 750 | 750 | 750 | 750 | 750 |
| Haute-Maurienne | 1910 | 2050 | 2480 | 2070 | 2270 | 2520 | 1050 | 1050 | 1050 | 1050 | 1050 | 1050 |
| Grandes-Rousses | 1660 | 1790 | 2170 | 1700 | 1940 | 2440 | 750 | 750 | 780 | 750 | 750 | 750 |
| Vercors | 1490 | 1700 | 1860 | 1580 | 1800 | 2150 | 620 | 850 | 1050 | 640 | 790 | 1020 |
| Oisans | 1690 | 1870 | 2230 | 1740 | 1970 | 2320 | 750 | 800 | 1030 | 750 | 750 | 770 |
| **Southern Alps** | | | | | | | | | | | | |
| Thabor | 1820 | 2040 | 2590 | 1850 | 2060 | 2470 | 1350 | 1350 | 1350 | 1350 | 1350 | 1350 |
| Pelvoux | 1630 | 2010 | 2690 | 1800 | 2020 | 2430 | 1050 | 1050 | 1050 | 1050 | 1050 | 1050 |
| Queyras | 2150 | 2480 | 2940 | 2210 | 2370 | 2790 | 1050 | 1050 | 1050 | 1050 | 1050 | 1050 |
| Devoluy | 1820 | 2030 | 2470 | 1840 | 2090 | 2470 | 780 | 1100 | 1280 | 750 | 900 | 1160 |
| Champsaur | 1680 | 1990 | 2540 | 1850 | 2080 | 2510 | 1050 | 1050 | 1050 | 1050 | 1050 | 1050 |
| Embrunnais Parpaillon | 2110 | 2520 | 2960 | 2040 | 2260 | 2810 | 750 | 940 | 1170 | 750 | 750 | 960 |





| | | | | | | | | | | | | |
|---|---|---|---|---|---|---|---|---|---|---|---|---|
| Ubaye | 2250 | 2560 | 2940 | 2300 | 2530 | 2910 | 1050 | 1050 | 1230 | 1050 | 1050 | 1050 |
| Haut-Var Haut-Verdon | 2140 | 2330 | 2580 | 2060 | 2280 | 2690 | 960 | 1230 | 1350 | 900 | 1080 | 1300 |
| Mercantour | 2210 | 2360 | 2760 | 2100 | 2330 | 2740 | 1050 | 1280 | 1360 | 1050 | 1240 | 1390 |
| **French Pyrenees** | | | | | | | | | | | | |
| Aspe Ossau | 1480 | 1620 | 1930 | 1740 | 1970 | 2210 | 960 | 1050 | 1400 | 1080 | 1220 | 1400 |
| Haute-Bigorre | 1670 | 1730 | 1950 | 1770 | 1980 | 2220 | 970 | 1060 | 1380 | 910 | 1080 | 1260 |
| Aure Louron | 1630 | 1860 | 1940 | 1830 | 2010 | 2270 | 930 | 990 | 1310 | 850 | 980 | 1210 |
| Luchonnais | 1650 | 1830 | 2020 | 1860 | 2020 | 2280 | 890 | 960 | 1420 | 750 | 900 | 1180 |
| Couserans | 1430 | 1620 | 1770 | 1560 | 1740 | 2050 | 900 | 1000 | 1180 | 800 | 930 | 1130 |
| Haute-Ariege | 1490 | 1640 | 1760 | 1690 | 1850 | 2140 | 890 | 970 | 1240 | 800 | 940 | 1130 |
| Orlu St-Barthelemy | 1580 | 1680 | 1800 | 1720 | 1880 | 2240 | 1050 | 1230 | 1310 | 1060 | 1170 | 1330 |
| Capcir Puymorens | 2050 | 2380 | 2580 | 2320 | 2570 | 2850 | 1200 | 1310 | 1540 | 1050 | 1260 | 1530 |
| Cerdagne Canigou | 2180 | 2360 | 2710 | 2320 | 2600 | 3000 | 1180 | 1310 | 1600 | 1180 | 1310 | 1550 |
| **Spanish and Andorran Pyrenees** | | | | | | | | | | | | |
| Andorra | 1790 | 1990 | 2370 | 1930 | 2100 | 2530 | 920 | 1130 | 1290 | 860 | 1080 | 1290 |
| Jacetiana | 1860 | 1930 | 2010 | 2010 | 2140 | 2340 | 1180 | 1280 | 1470 | 1170 | 1350 | 1530 |
| Gallego | 1780 | 1900 | 2040 | 2060 | 2240 | 2440 | 1110 | 1160 | 1360 | 1110 | 1240 | 1470 |
| Esera | 2050 | 2260 | 2360 | 2150 | 2290 | 2560 | 1040 | 1170 | 1590 | 840 | 1020 | 1340 |
| Aran | 1950 | 2070 | 2200 | 2080 | 2330 | 2630 | 1020 | 1140 | 1460 | 920 | 1110 | 1330 |
| Ribagorcana | 1960 | 2200 | 2350 | 2110 | 2300 | 2650 | 1070 | 1150 | 1340 | 890 | 1070 | 1340 |
| Pallaresa | 1900 | 2150 | 2450 | 2040 | 2260 | 2640 | 930 | 1050 | 1300 | 750 | 950 | 1200 |
| Ter-Freser | 2060 | 2570 | 2810 | 2060 | 2350 | 2780 | 1250 | 1320 | 1650 | 1130 | 1290 | 1440 |
| Cadi Moixero | 2010 | 2070 | 2450 | 2280 | 2530 | 2850 | 1060 | 1170 | 1460 | 980 | 1180 | 1380 |
| Pre-Pirineu | 1960 | 2060 | 2250 | 2190 | 2250 | 2250 | 1020 | 1230 | 1370 | 980 | 1180 | 1420 |



## A2 Near Future (2030 - 2050)

Table A2 Snow reliability elevation for the near future (2030 - 2050) for the 42 massifs distributed over the Northern Alps, Southern Alps, French Pyrenees and Spanish and Andorran Pyrenees, computed by climate models for the RCP 2.6, RCP 4.5 and RCP 8.5 and for three distinct reliability requirements (50%, 70% and 90%).

| | Groomed Snow | | | | | | | | | Including Snowmaking | | | | | | | | |
| | RCP 2.6 | | | RCP 4.5 | | | RCP 8.5 | | | RCP 2.6 | | | RCP 4.5 | | | RCP 8.5 | | |
| | (Quantiles) | | | (Quantiles) | | | (Quantiles) | | | (Quantiles) | | | (Quantiles) | | | (Quantiles) | | |
| Massif | 50% | 70% | 90% | 50% | 70% | 90% | 50% | 70% | 90% | 50% | 70% | 90% | 50% | 70% | 90% | 50% | 70% | 90% |
|---|---|---|---|---|---|---|---|---|---|---|---|---|---|---|---|---|---|---|
| **Northern Alps** | | | | | | | | | | | | | | | | | | |
| Chablais | 1710 | 1920 | 2150 | 1600 | 1860 | 2030 | 1680 | 1930 | 2260 | 530 | 800 | 880 | 690 | 850 | 990 | 680 | 840 | 1050 |
| Aravis | 1580 | 1800 | 2400 | 1510 | 1720 | 1990 | 1580 | 1800 | 2140 | 750 | 750 | 750 | 750 | 750 | 920 | 750 | 750 | 980 |
| Mont-Blanc | 1700 | 1900 | 2160 | 1510 | 1750 | 2030 | 1630 | 1830 | 2120 | 1050 | 1050 | 1050 | 1050 | 1050 | 1050 | 1050 | 1050 | 1050 |
| Bauges | 1720 | 1970 | 2180 | 1590 | 1860 | 2160 | 1580 | 1930 | 2250 | 450 | 710 | 940 | 450 | 750 | 1050 | 450 | 780 | 1050 |
| Beaufortain | 1620 | 1850 | 2330 | 1560 | 1750 | 2130 | 1630 | 1870 | 2280 | 750 | 750 | 750 | 750 | 750 | 770 | 750 | 750 | 930 |
| Haute-Tarentaise | 1780 | 2100 | 2490 | 1650 | 1850 | 2250 | 1730 | 2030 | 2420 | 750 | 750 | 750 | 750 | 750 | 750 | 750 | 750 | 750 |
| Chartreuse | 1870 | 2030 | 2250 | 1760 | 2020 | 2250 | 1840 | 2040 | 2250 | 770 | 880 | 1080 | 700 | 900 | 1150 | 720 | 930 | 1220 |
| Belledonne | 1660 | 1890 | 2120 | 1610 | 1840 | 2110 | 1690 | 1910 | 2260 | 610 | 720 | 950 | 640 | 780 | 1010 | 660 | 820 | 1080 |
| Maurienne | 1770 | 1990 | 2460 | 1640 | 1860 | 2160 | 1680 | 1930 | 2400 | 450 | 740 | 840 | 450 | 730 | 920 | 450 | 740 | 950 |
| Vanoise | 1740 | 2050 | 2490 | 1630 | 1840 | 2200 | 1740 | 1980 | 2400 | 750 | 750 | 750 | 750 | 750 | 750 | 750 | 750 | 750 |
| Haute-Maurienne | 2320 | 2470 | 2680 | 2200 | 2360 | 2670 | 2290 | 2460 | 2810 | 1050 | 1050 | 1050 | 1050 | 1050 | 1050 | 1050 | 1050 | 1050 |
| Grandes-Rousses | 1990 | 2220 | 2570 | 1910 | 2100 | 2540 | 1940 | 2210 | 2610 | 750 | 750 | 960 | 750 | 750 | 1010 | 750 | 750 | 1100 |
| Vercors | 1870 | 2030 | 2550 | 1910 | 2070 | 2400 | 1930 | 2140 | 2480 | 850 | 1020 | 1300 | 870 | 1030 | 1330 | 880 | 1080 | 1380 |
| Oisans | 1870 | 2180 | 2660 | 1940 | 2160 | 2560 | 2020 | 2280 | 2660 | 750 | 750 | 1020 | 750 | 750 | 1090 | 750 | 940 | 1200 |
| **Southern Alps** | | | | | | | | | | | | | | | | | | |
| Thabor | 2110 | 2410 | 2640 | 2050 | 2220 | 2580 | 2130 | 2440 | 2810 | 1350 | 1350 | 1350 | 1350 | 1350 | 1350 | 1350 | 1350 | 1350 |
| Pelvoux | 1990 | 2180 | 2900 | 1930 | 2190 | 2640 | 2050 | 2330 | 2820 | 1050 | 1050 | 1050 | 1050 | 1050 | 1050 | 1050 | 1050 | 1050 |
| Queyras | 2340 | 2620 | 3150 | 2320 | 2540 | 2900 | 2400 | 2690 | 3150 | 1050 | 1050 | 1050 | 1050 | 1050 | 1050 | 1050 | 1050 | 1050 |
| Devoluy | 2010 | 2250 | 2660 | 2100 | 2330 | 2650 | 2190 | 2430 | 2810 | 920 | 1150 | 1390 | 1000 | 1200 | 1430 | 1020 | 1270 | 1460 |
| Champsaur | 2010 | 2260 | 2790 | 2030 | 2310 | 2660 | 2190 | 2420 | 2880 | 1050 | 1050 | 1050 | 1050 | 1050 | 1050 | 1050 | 1050 | 1050 |
| Embrunnais Parpaillon | 2240 | 2480 | 3090 | 2190 | 2500 | 2950 | 2320 | 2640 | 3060 | 750 | 750 | 1020 | 750 | 960 | 1100 | 750 | 980 | 1160 |
| Ubaye | 2360 | 2860 | 3150 | 2460 | 2770 | 3120 | 2590 | 2870 | 3150 | 1050 | 1050 | 1330 | 1050 | 1050 | 1410 | 1050 | 1050 | 1560 |



| | | | | | | | | | | | | | | | | | | |
|---|---|---|---|---|---|---|---|---|---|---|---|---|---|---|---|---|---|---|
| Haut-Var Haut-Verdon | 2330 | 2620 | 2850 | 2300 | 2560 | 2850 | 2340 | 2650 | 2850 | 1140 | 1280 | 1500 | 1200 | 1350 | 1590 | 1200 | 1400 | 1690 |
| Mercantour | 2380 | 2640 | 3150 | 2350 | 2560 | 2880 | 2450 | 2680 | 3150 | 1250 | 1370 | 1560 | 1310 | 1430 | 1600 | 1330 | 1470 | 1690 |
| **French Pyrenees** | | | | | | | | | | | | | | | | | | |
| Aspe Ossau | 2000 | 2230 | 2480 | 2030 | 2280 | 2470 | 2070 | 2320 | 2590 | 1360 | 1500 | 1770 | 1380 | 1570 | 1870 | 1370 | 1570 | 2020 |
| Haute-Bigorre | 2070 | 2280 | 2690 | 2050 | 2230 | 2620 | 2080 | 2300 | 2820 | 1300 | 1480 | 1690 | 1220 | 1470 | 1710 | 1190 | 1460 | 1850 |
| Aure Louron | 2090 | 2340 | 2590 | 2050 | 2280 | 2590 | 2110 | 2370 | 2790 | 1260 | 1460 | 1700 | 1130 | 1350 | 1690 | 1140 | 1420 | 1870 |
| Luchonnais | 2070 | 2340 | 2590 | 2040 | 2270 | 2630 | 2080 | 2320 | 2660 | 1180 | 1400 | 1670 | 1140 | 1360 | 1700 | 1100 | 1380 | 1850 |
| Couserans | 1920 | 2120 | 2340 | 1830 | 2020 | 2330 | 1900 | 2100 | 2430 | 1120 | 1310 | 1520 | 1060 | 1220 | 1470 | 1110 | 1310 | 1600 |
| Haute-Ariege | 1960 | 2140 | 2380 | 1970 | 2080 | 2450 | 2010 | 2120 | 2560 | 1120 | 1300 | 1460 | 1070 | 1260 | 1470 | 1070 | 1270 | 1600 |
| Orlu St-Barthelemy | 2040 | 2280 | 2540 | 1950 | 2070 | 2540 | 2010 | 2200 | 2630 | 1280 | 1410 | 1630 | 1290 | 1430 | 1640 | 1280 | 1460 | 1750 |
| Capcir Puymorens | 2640 | 2850 | 2850 | 2580 | 2760 | 2850 | 2640 | 2820 | 2850 | 1450 | 1650 | 1860 | 1440 | 1600 | 1770 | 1400 | 1620 | 2040 |
| Cerdagne Canigou | 2610 | 2850 | 3150 | 2590 | 2860 | 3150 | 2640 | 2960 | 3150 | 1480 | 1630 | 1810 | 1450 | 1590 | 1800 | 1460 | 1650 | 1960 |
| **Spanish and Andorran Pyrenees** | | | | | | | | | | | | | | | | | | |
| Andorra | 2250 | 2550 | 2870 | 2140 | 2400 | 2820 | 2270 | 2580 | 3020 | 1290 | 1360 | 1540 | 1260 | 1390 | 1520 | 1290 | 1420 | 1720 |
| Jacetiana | 2280 | 2400 | 2600 | 2300 | 2380 | 2630 | 2320 | 2490 | 2850 | 1460 | 1680 | 1900 | 1550 | 1710 | 1920 | 1570 | 1760 | 1980 |
| Gallego | 2290 | 2410 | 2630 | 2310 | 2390 | 2650 | 2340 | 2600 | 3060 | 1450 | 1610 | 1800 | 1420 | 1630 | 1830 | 1460 | 1700 | 1980 |
| Esera | 2360 | 2620 | 2920 | 2360 | 2580 | 2830 | 2430 | 2660 | 3190 | 1310 | 1480 | 1700 | 1200 | 1450 | 1690 | 1320 | 1510 | 1930 |
| Aran | 2350 | 2660 | 3010 | 2340 | 2620 | 2960 | 2340 | 2610 | 3150 | 1260 | 1390 | 1700 | 1200 | 1370 | 1640 | 1230 | 1430 | 1830 |
| Ribagorcana | 2350 | 2590 | 2930 | 2380 | 2600 | 2870 | 2500 | 2670 | 3150 | 1350 | 1450 | 1770 | 1230 | 1420 | 1660 | 1300 | 1500 | 1880 |
| Pallaresa | 2350 | 2560 | 2870 | 2370 | 2590 | 2890 | 2450 | 2670 | 3140 | 1150 | 1320 | 1580 | 1140 | 1350 | 1580 | 1170 | 1430 | 1780 |
| Ter-Freser | 2370 | 2570 | 3000 | 2410 | 2690 | 3150 | 2510 | 2850 | 3150 | 1430 | 1530 | 1750 | 1410 | 1530 | 1740 | 1460 | 1570 | 1860 |
| Cadi Moixero | 2540 | 2850 | 2850 | 2600 | 2770 | 2850 | 2630 | 2850 | 2850 | 1350 | 1480 | 1750 | 1310 | 1490 | 1710 | 1380 | 1560 | 1920 |
| Pre-Pirineu | 2250 | 2250 | 2250 | 2250 | 2250 | 2250 | 2250 | 2250 | 2250 | 1310 | 1460 | 1780 | 1340 | 1500 | 1780 | 1360 | 1580 | 1950 |



## A3   End of the Century (2080 - 2100)

Table A3 Snow reliability elevation for the end of the century (2080 - 2100) for the 42 massifs distributed over the Northern Alps, Southern Alps, French Pyrenees and Spanish and Andorran Pyrenees, computed by climate models for the RCP 2.6, RCP 4.5 and RCP 8.5 and for three distinct reliability requirements (50%, 70% and 90%)

| | Groomed Snow | | | | | | | | | Including Snowmaking | | | | | | | | |
| | RCP 2.6 | | | RCP 4.5 | | | RCP 8.5 | | | RCP 2.6 | | | RCP 4.5 | | | RCP 8.5 | | |
| | (Quantiles) | | | (Quantiles) | | | (Quantiles) | | | (Quantiles) | | | (Quantiles) | | | (Quantiles) | | |
| **Massif** | 50% | 70% | 90% | 50% | 70% | 90% | 50% | 70% | 90% | 50% | 70% | 90% | 50% | 70% | 90% | 50% | 70% | 90% |
|---|---|---|---|---|---|---|---|---|---|---|---|---|---|---|---|---|---|---|
| **Northern Alps** | | | | | | | | | | | | | | | | | | |
| Chablais | 1640 | 1880 | 2330 | 1820 | 2060 | 2560 | 2380 | 2640 | 2850 | 630 | 790 | 930 | 750 | 900 | 1090 | 1100 | 1350 | 1980 |
| Aravis | 1490 | 1780 | 2290 | 1700 | 1950 | 2480 | 2330 | 2610 | 2850 | 750 | 750 | 750 | 750 | 750 | 1030 | 1010 | 1200 | 1850 |
| Mont-Blanc | 1570 | 1810 | 2390 | 1710 | 1950 | 2520 | 2310 | 2570 | 2900 | 1050 | 1050 | 1050 | 1050 | 1050 | 1050 | 1050 | 1050 | 1660 |
| Bauges | 1660 | 1980 | 2250 | 1780 | 2090 | 2250 | 2250 | 2250 | 2250 | 450 | 750 | 960 | 650 | 880 | 1240 | 1200 | 1640 | 2100 |
| Beaufortain | 1640 | 1820 | 2200 | 1730 | 1960 | 2600 | 2290 | 2570 | 3070 | 750 | 750 | 950 | 750 | 750 | 1000 | 1000 | 1130 | 1680 |
| Haute-Tarentaise | 1750 | 2160 | 2500 | 1820 | 2160 | 2820 | 2330 | 2670 | 3160 | 750 | 750 | 750 | 750 | 750 | 940 | 750 | 950 | 1140 |
| Chartreuse | 1830 | 2070 | 2250 | 2010 | 2250 | 2250 | 2250 | 2250 | 2250 | 700 | 950 | 1240 | 770 | 1060 | 1670 | 1790 | 1990 | 2220 |
| Belledonne | 1630 | 1800 | 2250 | 1800 | 2050 | 2560 | 2360 | 2610 | 3000 | 450 | 750 | 950 | 730 | 870 | 1190 | 1250 | 1620 | 2000 |
| Maurienne | 1780 | 1950 | 2370 | 1770 | 2090 | 2780 | 2330 | 2670 | 3120 | 450 | 690 | 980 | 680 | 810 | 1050 | 970 | 1170 | 1520 |
| Vanoise | 1720 | 2020 | 2470 | 1760 | 2140 | 2770 | 2320 | 2610 | 3080 | 750 | 750 | 750 | 750 | 750 | 750 | 750 | 1090 | 1440 |
| Haute-Maurienne | 2320 | 2500 | 2800 | 2360 | 2610 | 2930 | 2710 | 2970 | 3360 | 1050 | 1050 | 1050 | 1050 | 1050 | 1050 | 1050 | 1050 | 1500 |
| Grandes-Rousses | 1930 | 2210 | 2560 | 2010 | 2380 | 2810 | 2570 | 2850 | 3240 | 750 | 750 | 1000 | 750 | 750 | 1160 | 1120 | 1390 | 1830 |
| Vercors | 1830 | 2130 | 2550 | 2050 | 2310 | 2550 | 2550 | 2550 | 2550 | 920 | 1100 | 1300 | 980 | 1310 | 1660 | 1750 | 2020 | 2360 |
| Oisans | 1990 | 2160 | 2800 | 2090 | 2430 | 2900 | 2630 | 2890 | 3430 | 750 | 750 | 1000 | 750 | 1020 | 1290 | 1220 | 1580 | 1940 |
| **Southern Alps** | | | | | | | | | | | | | | | | | | |
| Thabor | 2150 | 2400 | 2720 | 2250 | 2570 | 2900 | 2790 | 2990 | 3150 | 1350 | 1350 | 1350 | 1350 | 1350 | 1350 | 1350 | 1350 | 1740 |
| Pelvoux | 2110 | 2340 | 2850 | 2140 | 2450 | 2900 | 2660 | 2960 | 3400 | 1050 | 1050 | 1050 | 1050 | 1050 | 1050 | 1050 | 1050 | 1410 |
| Queyras | 2460 | 2690 | 3150 | 2610 | 2900 | 3150 | 2980 | 3150 | 3150 | 1050 | 1050 | 1050 | 1050 | 1050 | 1050 | 1050 | 1240 | 1480 |
| Devoluy | 2070 | 2330 | 2640 | 2330 | 2570 | 2880 | 2880 | 3060 | 3150 | 1050 | 1230 | 1380 | 1160 | 1320 | 1700 | 1730 | 2000 | 2330 |
| Champsaur | 2180 | 2440 | 2820 | 2330 | 2560 | 3020 | 2820 | 3120 | 3450 | 1050 | 1050 | 1050 | 1050 | 1050 | 1050 | 1050 | 1350 | 1710 |
| Embrunnais Parpaillon | 2430 | 2640 | 3060 | 2510 | 2820 | 3220 | 2970 | 3220 | 3450 | 750 | 750 | 1020 | 750 | 1060 | 1210 | 1300 | 1620 | 2130 |
| Ubaye | 2610 | 2890 | 3150 | 2760 | 2970 | 3150 | 3030 | 3150 | 3150 | 1050 | 1050 | 1300 | 1050 | 1250 | 1660 | 1560 | 1880 | 2320 |





| | | | | | | | | | | | | | | | | | | |
|---|---|---|---|---|---|---|---|---|---|---|---|---|---|---|---|---|---|---|
| Haut-Var Haut-Verdon | 2330 | 2560 | 2850 | 2540 | 2780 | 2850 | 2850 | 2850 | 2850 | 1160 | 1330 | 1560 | 1300 | 1540 | 1850 | 1900 | 2090 | 2410 |
| Mercantour | 2300 | 2570 | 2880 | 2590 | 2820 | 3150 | 3020 | 3150 | 3150 | 1270 | 1340 | 1510 | 1370 | 1500 | 1760 | 1750 | 1990 | 2480 |
| **French Pyrenees** | | | | | | | | | | | | | | | | | | |
| Aspe Ossau | 2000 | 2190 | 2540 | 2260 | 2400 | 2650 | 2780 | 2940 | 3150 | 1350 | 1470 | 1840 | 1480 | 1750 | 2130 | 2180 | 2380 | 2840 |
| Haute-Bigorre | 2000 | 2180 | 2480 | 2250 | 2380 | 2980 | 2990 | 3200 | 3450 | 1290 | 1470 | 1610 | 1400 | 1620 | 1890 | 1990 | 2220 | 3060 |
| Aure Louron | 2030 | 2190 | 2650 | 2310 | 2490 | 2880 | 2900 | 3150 | 3150 | 1130 | 1300 | 1720 | 1340 | 1670 | 1960 | 1980 | 2180 | 2720 |
| Luchonnais | 2000 | 2220 | 2740 | 2280 | 2550 | 2860 | 2900 | 3150 | 3450 | 1160 | 1360 | 1750 | 1330 | 1600 | 1950 | 1990 | 2180 | 2680 |
| Couserans | 1840 | 2030 | 2350 | 2030 | 2260 | 2480 | 2600 | 2830 | 3150 | 1060 | 1230 | 1490 | 1250 | 1400 | 1700 | 1830 | 2020 | 2560 |
| Haute-Ariege | 1910 | 2050 | 2410 | 2060 | 2330 | 2790 | 2770 | 3010 | 3150 | 1050 | 1220 | 1480 | 1180 | 1410 | 1690 | 1760 | 2020 | 2590 |
| Orlu St-Barthelemy | 1910 | 2140 | 2370 | 2100 | 2440 | 2840 | 2690 | 2850 | 2850 | 1260 | 1410 | 1630 | 1380 | 1570 | 1850 | 1900 | 2140 | 2680 |
| Capcir Puymorens | 2510 | 2690 | 2850 | 2710 | 2850 | 2850 | 2850 | 2850 | 2850 | 1430 | 1540 | 1800 | 1530 | 1740 | 2110 | 2170 | 2370 | 2770 |
| Cerdagne Canigou | 2450 | 2640 | 3150 | 2770 | 3060 | 3150 | 3150 | 3150 | 3150 | 1470 | 1620 | 1830 | 1550 | 1730 | 2020 | 2010 | 2270 | 2750 |
| **Spanish and Andorran Pyrenees** | | | | | | | | | | | | | | | | | | |
| Andorra | 2090 | 2240 | 2560 | 2360 | 2750 | 3150 | 3010 | 3150 | 3150 | 1230 | 1350 | 1600 | 1350 | 1490 | 1810 | 1870 | 2070 | 2640 |
| Jacetiana | 2280 | 2410 | 2700 | 2360 | 2570 | 2810 | 2950 | 3150 | 3150 | 1470 | 1680 | 1910 | 1690 | 1870 | 2060 | 2190 | 2370 | 2810 |
| Gallego | 2350 | 2510 | 2750 | 2390 | 2640 | 2990 | 3040 | 3150 | 3150 | 1390 | 1500 | 1730 | 1590 | 1780 | 2030 | 2140 | 2390 | 2880 |
| Esera | 2440 | 2610 | 2950 | 2570 | 2790 | 3080 | 3060 | 3260 | 3450 | 1300 | 1500 | 1640 | 1460 | 1670 | 1940 | 2050 | 2230 | 2720 |
| Aran | 2260 | 2520 | 2780 | 2570 | 2850 | 3150 | 3120 | 3150 | 3150 | 1120 | 1340 | 1620 | 1340 | 1550 | 1970 | 1940 | 2160 | 2720 |
| Ribagorcana | 2350 | 2680 | 2950 | 2590 | 2820 | 3150 | 3040 | 3150 | 3150 | 1300 | 1500 | 1710 | 1460 | 1650 | 1910 | 2040 | 2250 | 2690 |
| Pallaresa | 2350 | 2620 | 3020 | 2630 | 2840 | 3150 | 3130 | 3150 | 3150 | 1170 | 1340 | 1710 | 1390 | 1570 | 1880 | 2010 | 2200 | 2710 |
| Ter-Freser | 2320 | 2600 | 3040 | 2670 | 2920 | 3150 | 3150 | 3150 | 3150 | 1370 | 1480 | 1730 | 1510 | 1670 | 1950 | 2020 | 2280 | 2740 |
| Cadi Moixero | 2500 | 2850 | 2850 | 2730 | 2850 | 2850 | 2850 | 2850 | 2850 | 1350 | 1550 | 1800 | 1480 | 1690 | 2010 | 1990 | 2290 | 2850 |
| Pre-Pirineu | 2250 | 2250 | 2250 | 2250 | 2250 | 2250 | 2250 | 2250 | 2250 | 1210 | 1500 | 1800 | 1470 | 1710 | 1970 | 1970 | 2210 | 2250 |

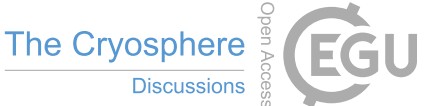



## Appendix B: Detailed features of individual ski resorts

Table B Main features of the 175 ski resorts included in the present work grouped by massifs and major areas (Northern and Southern Alps, French and Spanish and Andorran Pyrenees).

| | Ski Lift Power | Size Category | Village Elevation | Mean Elevation | Max. Elevation |
|---|---|---|---|---|---|
| **Chablais** (Northern Alps) | | | | | |
| LULLIN COL DE FEU | 81 | S | 1084 | 1130 | 1175 |
| PLAINE-JOUX | 749 | S | 1372 | 1508 | 1718 |
| ABONDANCE | 1205 | S | 1049 | 1341 | 1758 |
| HABERE POCHE | 1454 | S | 1018 | 1200 | 1505 |
| BELLEVAUX HIRMENTAZ | 2115 | S | 1185 | 1331 | 1612 |
| BERNEX | 2372 | S | 1009 | 1396 | 1871 |
| THOLLON LES MEMISES | 2468 | S | 1048 | 1518 | 1938 |
| BRASSES (LES) | 2617 | M | 1148 | 1249 | 1495 |
| ESPACE ROC D'ENFER | 3100 | M | 1013 | 1351 | 1790 |
| CHAPELLE D'ABONDANCE (LA) | 3156 | M | 1054 | 1410 | 1797 |
| PRAZ-DE-LYS - SOMMAND | 5099 | L | 1453 | 1487 | 1961 |
| CARROZ D'ARACHES (LES) | 7348 | L | 1160 | 1561 | 2109 |
| MORZINE PLENEY NYON | 9204 | L | 1012 | 1467 | 2127 |
| GETS (LES) | 10489 | L | 1202 | 1502 | 2131 |
| MORILLON-SAMOENS-SIXT | 12159 | L | 968 | 1501 | 2118 |
| GRAND MASSIF (FLAINE - VALLEE DU GIFFRE) | 13466 | L | 1662 | 1982 | 2482 |
| CHATEL | 13959 | L | 1208 | 1631 | 2093 |
| AVORIAZ - MORZINE | 18826 | XL | 1758 | 1815 | 2501 |
| **Aravis** (Northern Alps) | | | | | |





| | | | | | |
|---|---|---|---|---|---|
| CRET (SAINT-JEAN-DE-SIXT) | 49 | S | 959 | 843 | 1020 |
| MONTMIN | 96 | S | 1152 | 1101 | 1195 |
| REPOSOIR (LE) | 271 | S | 1039 | 1301 | 1626 |
| RAFFORTS (LES) - UGINE | 285 | S | 939 | 1067 | 1225 |
| NANCY SUR CLUSES | 354 | S | 1291 | 1341 | 1558 |
| MONT SAXONNEX | 828 | S | 1059 | 1346 | 1574 |
| PORTES DU MONT BLANC (LES) - SALLANCHE-CORDON | 1005 | S | 1106 | 1315 | 1538 |
| MANIGOD CROIX FRY | 2088 | S | 1502 | 1491 | 1795 |
| PORTES DU MONT BLANC (LES) - COMBLOUX - LE JAILLET - LA GIETTAZ | 4753 | M | 1152 | 1405 | 1982 |
| GRAND BORNAND (LE) | 11400 | L | 1254 | 1509 | 2031 |
| CLUSAZ (LA) | 13826 | L | 1126 | 1612 | 2375 |
| **Mont-Blanc** (Northern Alps) | | | | | |
| VALLORCINE LA POYA | 1503 | S | 1358 | 1577 | 1932 |
| SAINT NICOLAS DE VEROCE | 3657 | M | 1241 | 1751 | 2364 |
| LES HOUCHES - SAINT-GERVAIS | 5872 | L | 1068 | 1532 | 1892 |
| SAINT GERVAIS BETTEX | 7293 | L | 1084 | 1549 | 2386 |
| CONTAMINES (LES)-HAUTELUCE | 10409 | L | 1206 | 1786 | 2437 |
| MEGEVE | 15132 | XL | 1175 | 1557 | 2014 |
| CHAMONIX | 27378 | XL | 1160 | 1938 | 3787 |
| **Bauges** (Northern Alps) | | | | | |
| SEYTHENEX - LA SAMBUY | 1170 | S | 1160 | 1429 | 1835 |
| SAVOIE GRAND REVARD | 1287 | S | 1376 | 1339 | 1549 |
| SEMNOZ (LE) | 1474 | S | 1480 | 1505 | 1696 |
| AILLON LE JEUNE-MARGERIAZ | 3594 | M | 1029 | 1430 | 1834 |
| **Beaufortain** (Northern Alps) | | | | | |
| GRANIER SUR AIME | 224 | S | 1394 | 1522 | 1661 |
| CREST VOLAND | 3472 | M | 1257 | 1410 | 1608 |





| | | | | | |
|---|---|---|---|---|---|
| ARECHES BEAUFORT | 4247 | M | 1104 | 1573 | 2137 |
| VAL D'ARLY | 8345 | L | 1158 | 1498 | 2053 |
| SAISIES (LES) | 8433 | L | 1529 | 1727 | 2052 |
| **Haute-Tarentaise** (Northern Alps) | | | | | |
| SAINTE FOY TARENTAISE | 2436 | S | 1536 | 2067 | 2612 |
| ROSIERE (LA) | 6969 | L | 1841 | 2031 | 2572 |
| VAL D'ISERE | 24371 | XL | 1868 | 2368 | 3197 |
| TIGNES | 25814 | XL | 2092 | 2251 | 3459 |
| ARCS (LES) - PEISEY-VALLANDRY | 31699 | XL | 1786 | 1826 | 3220 |
| **Chartreuse** (Northern Alps) | | | | | |
| COL DE MARCIEU | 221 | S | 1070 | 1184 | 1350 |
| SAPPEY EN CHARTREUSE (LE) | 362 | S | 988 | 1104 | 1344 |
| COL DE PORTE | 372 | S | 1329 | 1370 | 1615 |
| COL DU GRANIER - DESERT D'ENTREMONT (LE) | 506 | S | 1106 | 1207 | 1428 |
| SAINT HILAIRE DU TOUVET | 517 | S | 974 | 1075 | 1415 |
| SAINT PIERRE DE CHARTREUSE - LE PLANOLET | 2958 | M | 982 | 1318 | 1751 |
| **Belledonne** (Northern Alps) | | | | | |
| COL DU BARIOZ ALPIN | 190 | S | 1366 | 1505 | 1684 |
| COLLET D'ALLEVARD (LE) | 2897 | M | 1452 | 1715 | 2091 |
| CHAMROUSSE | 7078 | L | 1732 | 1880 | 2253 |
| SEPT LAUX (LES) | 10881 | L | 1396 | 1786 | 2378 |
| **Maurienne** (Northern Alps) | | | | | |
| SAINT-COLOMBAN-DES-VILLARDS | 1732 | S | 1117 | 1586 | 2234 |
| ALBIEZ MONTROND | 2708 | M | 1570 | 1725 | 2060 |
| KARELLIS (LES) | 4986 | M | 1608 | 2043 | 2490 |
| TOUSSUIRE (LA) - SAINT-PANCRACE (LES BOTTIERES) | 6148 | L | 1667 | 1939 | 2367 |
| CORBIER (LE)-SAINT JEAN D'ARVES | 6363 | L | 1555 | 1791 | 2377 |





| | | | | | |
|---|---|---|---|---|---|
| VALMEINIER | | 7718 | L | 1719 | 2017 | 2579 |
| VALLOIRE | | 9631 | L | 1482 | 1597 | 2530 |

**Vanoise** (Northern Alps)

| | | | | | |
|---|---|---|---|---|---|
| NOTRE DAME DU PRE | 226 | S | 1279 | 1365 | 1510 |
| AUSSOIS | 3055 | M | 1535 | 2096 | 2670 |
| PRALOGNAN | 3505 | M | 1438 | 1495 | 2340 |
| ORELLE | 5217 | L | 2364 | 2003 | 3242 |
| SAINT FRANCOIS LONGCHAMP | 6405 | L | 1583 | 1904 | 2514 |
| VALMOREL | 11005 | L | 1382 | 1748 | 2401 |
| MERIBEL LES ALLUES | 15767 | XL | 1362 | 1913 | 2701 |
| VAL THORENS | 19844 | XL | 2300 | 2501 | 3186 |
| MENUIRES (LES) | 22331 | XL | 1798 | 2185 | 2845 |
| PLAGNE (LA) | 35044 | XL | 1849 | 2028 | 3167 |
| COURCHEVEL | 39787 | XL | 1667 | 2084 | 2919 |

**Haute-Maurienne** (Northern Alps)

| | | | | | |
|---|---|---|---|---|---|
| BRAMANS | 16 | S | 1261 | 1277 | 1315 |
| BESSANS | 185 | S | 1715 | 1849 | 2079 |
| BONNEVAL SUR ARC | 2024 | S | 1831 | 2339 | 2937 |
| VAL FREJUS | 3773 | M | 1627 | 2086 | 2731 |
| NORMA (LA) | 4032 | M | 1387 | 1964 | 2742 |
| VAL CENIS | 13212 | L | 1440 | 1921 | 2737 |

**Grandes-Rousses** (Northern Alps)

| | | | | | |
|---|---|---|---|---|---|
| CHAZELET-VILLAR D'ARENE | 1088 | S | 1664 | 1898 | 2164 |
| SAINT SORLIN D'ARVES | 7746 | L | 1556 | 2028 | 2590 |
| OZ - VAUJANY | 8072 | L | 1311 | 1853 | 2817 |
| ALPE D'HUEZ (L') | 18232 | XL | 1771 | 2125 | 3318 |

**Vercors** (Northern Alps)





| | | | | | |
|---|---|---|---|---|---|
| SAINT NIZIER | 22 | S | 1176 | 1181 | 1200 |
| RENCUREL | 221 | S | 1081 | 1137 | 1233 |
| COL DE L'ARZELIER | 472 | S | 1171 | 1311 | 1477 |
| FONT D'URLE - CHAUD CLAPIER | 504 | S | 1433 | 1405 | 1542 |
| GRESSE EN VERCORS | 1257 | S | 1251 | 1396 | 1703 |
| COL DU ROUSSET | 1297 | S | 1275 | 1424 | 1695 |
| AUTRANS | 1535 | S | 1074 | 1415 | 1650 |
| MEAUDRE | 1645 | S | 1009 | 1265 | 1577 |
| LANS EN VERCORS | 1880 | S | 1137 | 1523 | 1801 |
| VILLARD DE LANS-CORRENCON | 9644 | L | 1221 | 1575 | 2052 |

**Oisans** (Northern Alps)

| | | | | | |
|---|---|---|---|---|---|
| NOTRE DAME DE VAULX | 18 | S | 972 | 1058 | 1085 |
| VILLARD REYMOND | 37 | S | 1650 | 1691 | 1712 |
| MOTTE D'AVEILLANS (LA) | 84 | S | 1285 | 1360 | 1430 |
| SAINT FIRMIN VALGAUDEMAR | 91 | S | 1306 | 1470 | 1580 |
| COL D'ORNON | 401 | S | 1366 | 1559 | 1855 |
| GRAVE (LA) | 995 | S | 1498 | 2479 | 3532 |
| ALPE DU GRAND SERRE (L') | 3225 | M | 1403 | 1716 | 2221 |
| DEUX ALPES (LES) | 23796 | XL | 1720 | 2344 | 3642 |

**Thabor** (Southern Alps)

| | | | | | |
|---|---|---|---|---|---|
| NEVACHE | 112 | S | 1609 | 1643 | 1707 |
| MONTGENEVRE | 8587 | L | 1845 | 2143 | 2581 |

**Pelvoux** (Southern Alps)

| | | | | | |
|---|---|---|---|---|---|
| PELVOUX-VALLOUISE | 1391 | S | 1398 | 1615 | 2237 |
| PUY ST VINCENT | 5734 | L | 1645 | 1938 | 2668 |
| SERRE CHEVALIER | 26571 | XL | 1376 | 1993 | 2750 |

**Queyras** (Southern Alps)





| STATION DU QUEYRAS | 6834 | L | 1819 | 2024 | 2801 |
|---|---|---|---|---|---|
| **Devoluy** (Southern Alps) | | | | | |
| LUS LA JARJATTE | 385 | S | 1171 | 1339 | 1521 |
| MASSIF DU DEVOLUY | 7068 | L | 1506 | 1591 | 2490 |
| **Champsaur** (Southern Alps) | | | | | |
| ANCELLE | 1842 | S | 1351 | 1511 | 1811 |
| STATIONS VILLAGE DU CHAMPSAUR | 3907 | M | 1386 | 1486 | 2240 |
| ORCIERES MERLETTE | 8297 | L | 1836 | 2178 | 2725 |
| **Embrunnais Parpaillon** (Southern Alps) | | | | | |
| REALLON | 1408 | S | 1569 | 1789 | 2114 |
| ORRES (LES) | 6545 | L | 1687 | 2027 | 2704 |
| RISOUL | 6734 | L | 1900 | 2188 | 2551 |
| **Ubaye** (Southern Alps) | | | | | |
| COL SAINT JEAN | 2952 | M | 1345 | 1883 | 2450 |
| STATIONS DE L'UBAYE | 5825 | L | 1523 | 1909 | 2427 |
| PRA-LOUP | 6772 | L | 1621 | 1904 | 2500 |
| VARS | 9073 | L | 1832 | 2079 | 2721 |
| **Haut-Var Haut-Verdon** (Southern Alps) | | | | | |
| VAL PELENS | 169 | S | 1612 | 1662 | 1737 |
| ROUBION LES BUISSES | 728 | S | 1443 | 1611 | 1898 |
| VALBERG-BEUIL | 4849 | M | 1665 | 1650 | 2020 |
| VAL D'ALLOS | 8257 | L | 1730 | 1580 | 2500 |
| **Mercantour** (Southern Alps) | | | | | |
| STATIONS DU MERCANTOUR | 17669 | XL | 1784 | 2029 | 2585 |
| **Aspe Ossau** (French Pyrenees) | | | | | |
| ARTOUSTE | 2565 | M | 1894 | 1730 | 2040 |
| GOURETTE - PIERRE SAINT MARTIN (LA) | 8788 | L | 1420 | 1543 | 2453 |



**Haute-Bigorre** (French Pyrenees)

| | | | | | |
|---|---|---|---|---|---|
| VAL D'AZUN | 14 | S | 1469 | 1469 | 1469 |
| PIC DU MIDI | 516 | S | 1780 | 2292 | 2856 |
| HAUTACAM | 919 | S | 1520 | 1454 | 1729 |
| GAVARNIE | 1999 | S | 1846 | 1997 | 2282 |
| PIAU ENGALY | 3819 | M | 1841 | 2030 | 2529 |
| LUZ ARDIDEN | 4099 | M | 1716 | 1951 | 2484 |
| CAUTERETS | 7193 | L | 1755 | 1932 | 2416 |
| TOURMALET | 10243 | L | 1784 | 1866 | 2490 |
| SAINT LARY SOULAN | 12822 | L | 1653 | 1991 | 2471 |

**Aure Louron** (French Pyrenees)

| | | | | | |
|---|---|---|---|---|---|
| VAL LOURON | 1693 | S | 1462 | 1723 | 2058 |
| PEYRAGUDES | 7741 | L | 1623 | 1884 | 2260 |

**Luchonnais** (French Pyrenees)

| | | | | | |
|---|---|---|---|---|---|
| BOURG D'OUEIL | 109 | S | 1345 | 1438 | 1498 |
| SUPERBAGNERES | 6446 | L | 1792 | 1736 | 2133 |

**Couserans** (French Pyrenees)

| | | | | | |
|---|---|---|---|---|---|
| LE MOURTIS | 1096 | S | 1425 | 1578 | 1801 |
| GUZET NEIGE | 2673 | M | 1445 | 1600 | 2050 |

**Haute-Ariege** (French Pyrenees)

| | | | | | |
|---|---|---|---|---|---|
| AX LES THERMES | 7437 | L | 1398 | 1955 | 2948 |

**Orlu St-Barthelemy** (French Pyrenees)

| | | | | | |
|---|---|---|---|---|---|
| CAMURAC | 527 | S | 1417 | 1335 | 1755 |
| ASCOU | 820 | S | 1558 | 1731 | 2058 |
| MIJANE - GOULIER - PLATEAU DE BEILLE | 891 | S | 1663 | 1599 | 2013 |
| MONTS D'OLMES | 1922 | S | 1487 | 1647 | 1948 |

**Capcir Puymorens** (French Pyrenees)





| | | | | | |
|---|---|---|---|---|---|
| QUILLANE (LA) | 111 | S | 1709 | 1752 | 1812 |
| PORTE PUYMORENS | 1800 | S | 1755 | 1259 | 2342 |
| FORMIGUERES | 1869 | S | 1769 | 1974 | 2320 |
| FONT ROMEU - P2000 | 5132 | L | 1775 | 1982 | 2227 |
| ANGLES (LES) | 5478 | L | 1683 | 1968 | 2361 |
| **Cerdagne Canigou** (French Pyrenees) | | | | | |
| CAMBRE D'AZE | 1741 | S | 1745 | 1958 | 2424 |
| **Andorra** (Spanish and Andorran Pyrenees) | | | | | |
| ARINSAL | 1663 | S | 1706 | 2147 | 2531 |
| PAL | 3054 | M | 1651 | 2062 | 2351 |
| ORDINO-ARCALIS | 3897 | M | 1792 | 2281 | 2633 |
| GRANDVALIRA | 19747 | XL | 1772 | 2251 | 2669 |
| **Jacetiana** (Spanish and Andorran Pyrenees) | | | | | |
| ASTUN | 3304 | M | 1591 | 1968 | 2249 |
| CANDANCHU | 4573 | M | 1506 | 1836 | 2283 |
| FORMIGAL | 11251 | L | 1562 | 1923 | 2263 |
| **Gallego** (Spanish and Andorran Pyrenees) | | | | | |
| PANTICOSA | 2799 | M | 1476 | 1789 | 2191 |
| **Esera** (Spanish and Andorran Pyrenees) | | | | | |
| BENASQUE | 7000 | L | 1694 | 2129 | 2645 |
| **Aran** (Spanish and Andorran Pyrenees) | | | | | |
| TAVASCAN | 774 | S | 1582 | 1954 | 2220 |
| BAQUEIRA BERET | 21246 | XL | 1685 | 2115 | 2543 |
| **Ribagorcana** (Spanish and Andorran Pyrenees) | | | | | |
| BOI TAULL | 4648 | M | 1825 | 2333 | 2741 |
| **Pallaresa** (Spanish and Andorran Pyrenees) | | | | | |
| ESPOT | 2554 | M | 1609 | 1997 | 2339 |





| PORT AINE | | 2927 | M | 1714 | 2160 | 2432 |
|---|---|---|---|---|---|---|
| **Ter-Freser** (Spanish and Andorran Pyrenees) | | | | | | |
| VALL DE NURIA | | 1040 | S | 1656 | 2070 | 2303 |
| VALLTER 2000 | | 2036 | S | 1797 | 2289 | 2526 |
| **Cadi Moixero** (Spanish and Andorran Pyrenees) | | | | | | |
| LA MOLINA | | 14282 | L | 1603 | 1988 | 2527 |
| **Pre-Pirineu** (Spanish and Andorran Pyrenees) | | | | | | |
| PORT DEL COMTE | | 5301 | L | 1624 | 2020 | 2329 |