# Peer review of "Winter tourism under climate change in the Pyrenees and the French Alps: relevance of snowmaking as a technical adaptation"

_The Cryosphere, 2018_

## Referee Comment (RC1) · Demiroglu (Referee) · 16 Jan 2019

The paper fulfills a major spatial gap in the ski tourism and climate change literature. Two minor corrections are needed before its publication:

1- Snowmaking is referred as a "mitigation method" throughout the paper (p.2 line 34, p.8 line 12). This should be corrected as "adaptation method" in line with the common literature and the IPCC terminology. "Mitigating the impacts" phrases (p. 1 line 18, p.9 lines 4&6, p.15 line 2) could also be rephrased (to e.g. "reducing") to avoid any confusion.

[Figure]

2- p.2 line 15. "Slovenia" should be replaced with "Switzerland".

Thanks for your efforts and looking forward.

---

## Referee Comment (RC2) · Anonymous Referee #2 · 17 Jan 2019

The authors present a very interesting and innovative study on the effects of climate change on winter tourism in the Pyrenees and French Alps. The presented manuscript goes one step beyond most other publications in this field, by including technical snow in a detailed way on a larger scale. The modeling approach behind the study is cutting edge, no further comments needed here. Several simplifications in the definition of parameters related to technical snow production, as used in this study, cannot be avoided in such a large-scale overview analysis.

The only major critical comment I have is, that on the one hand economic analysis is excluded and adaptation options are restricted to snow-making and grooming, but on the

other hand the authors interpret the results of the study in terms like skiing resorts being "at risk". This kind of interpretation should be avoided, since the risk is in this case an economic risk, which cannot be analyzed by a pure scientific-technical study, which additionally lacks a comprehensive analysis of adaptation options. Tourist resorts have many options to adapt to new conditions, not only snow-making and grooming. A study like this (on snow reliability) can be very valuable for tourist resorts as background information for developing long-term strategies, but it cannot conclude about the risk the resort is at. Therefore, the authors should rephrase the interpretation of their results and be careful with the term "risk". However, this is only a minor revision, since it affects only the phrasing of a few sentences in the manuscript (in the abstract, section 3.2 and conclusions).

Minor comment: As the other reviewer pointed out, the term "mitigation" should be strictly avoided in this context, since mitigation is commonly use to denote activities that aim to avoid or minimize climate change. Technical snowmaking can rather be called an adaptation to climate change.

---

## Referee Comment (RC3) · Robert Steiger (Referee) · 22 Feb 2019

General comments In this paper results from snowpack modeling tailored to ski resort operations and potential impacts of climate change are presented. From a regional perspective it is one of the view climate impact analyses for the ski industry in France, one of the most important ski markets in the world which has so far been under-researched. The applied model was already introduced elsewhere, but to my knowledge this is the first application for assessing the future perspectives of the ski industry. It is an important contribution as snowmaking - an important adaptation strategy - was included. I have some question in the methods section (see comments below) and I suggest to in-

[Figure]

clude a paragraph in the discussion section on the "take-home message" of the paper. Apart from % changes illustrated in the results section, is it possible to evaluate the (near or longer term) future for skiing tourism in France and the Spanish Pyrenées? Is climate change a serious challenge, or manageable?

Specific comments p. 3, l. 2: Damm et al. (2017) did not include snowmaking in their assessment, so the reference does not match your statement ". . .and the snowmaking requirements so as to compensate the loss over Europe (Damm. . .). p.4, l.5: what is the justification for using the village elevation? p. 4, l.7-15: It is not clear to me what data was available in which region and which data you had to estimate. I understand that all data (village elevation, min/max elev., ski lift power and surface area) was available for France. As seen in Fig 1 you then estimated ski lift power (?) based on the surface area you drew from OSM and on the linear model derived from French ski resorts? Then you also had to estimate the elevations of Spanish/Andorran ski areas? Why that if you had OSM data? How can you explain the outliers in the OSM/BD stations figure? p. 5, l. 12: can you add some data on Spain as well? (the ski areas in this study represent xy% of ski lift infrastructures of Spain) p. 6, l. 11: 150 kg/m$^2$ -> if it is density it should be kg/m3 ; this is an uncommon density for technical snow for base layer snowmaking, typically it is around 400 kg/m$^3$ p. 8, l. 21-23: please explain why the village elevation is relevant in your assessment. Later on you refer to the "lowest elevation of the ski area", this would be a clear explanation p. 9, l. 2. "snowmaking is limited to the lowest elevation and for a minority of seasons" -> I don't understand this sentence. How is snowmaking limited to a minority of seasons? Does that mean that snow is only produced in some years? p. 11, l. 4: "to decrease in the Pyrenees, up to 15%" -> here the sentence structure confused me a bit because "decrease" is followed by a positive number and in the same sentence there is 15% another time, but as increase. Maybe consider to split this sentence in two? p. 11, l. 5: what do you mean by "either in the Northern or Southern Alps"? Fig. 4/Discussion: how can you explain that the systematic bias is not existent in the Southern Alps?

---

## Author Comment (AC1) · 17 Mar 2019

*We thank O. Cenk Demiroglu for useful feedback.*

**1   Specific comments**

Snowmaking is referred as a "mitigation method" throughout the paper (p.2 line 34, p.8 line 12). This should be corrected as "adaptation method" in line with the common literature and the IPCC terminology.  "Mitigating the impacts" phrases (p. 1 line 18,

p.9 lines 4&6, p.15 line 2) could also be rephrased (to e.g. "reducing") to avoid any confusion.

*We agree with the suggestion. The term "mitigation" was replaced with the term "adaptation" where appropriate.*

p.2 line 15. "Slovenia" should be replaced with "Switzerland".

*We agree and have replaced "Slovenia" with "Switzerland".*

---

## Author Comment (AC2) · 17 Mar 2019

**1   General comments**

The authors present a very interesting and innovative study on the effects of climate change on winter tourism in the Pyrenees and French Alps. The presented manuscript goes one step beyond most other publications in this field, by including technical snow in a detailed way on a larger scale. The modeling approach behind the study is cutting edge, no further comments needed here.  Several simplifications in the definition of parameters related to technical snow production, as used in this study, cannot be

avoided in such a large-scale overview analysis.

*We thank the reviewer for constructive and useful feedback. Please find below some detailed replies on the specific comments:*

**2 Specific comments**

The only major critical comment I have is, that on the one hand economic analysis is excluded and adaptation options are restricted to snow-making and grooming, but on the other hand the authors interpret the results of the study in terms like skiing resorts being "at risk". This kind of interpretation should be avoided, since the risk is in this case an economic risk, which cannot be analyzed by a pure scientific-technical study, which additionally lacks a comprehensive analysis of adaptation options. Tourist resorts have many options to adapt to new conditions, not only snow-making and grooming. A study like this (on snow reliability) can be very valuable for tourist resorts as background information for developing long-term strategies, but it cannot conclude about the risk the resort is at. Therefore, the authors should rephrase the interpretation of their results and be careful with the term "risk". However, this is only a minor revision, since it affects only the phrasing of a few sentences in the manuscript (in the abstract, section 3.2 and conclusions).

*We agree that the term risk, in the climate change literature, and in IPCC assessments, encompasses the three dimensions of climate change hazard, vulnerability and exposure. Here we mainly focus on the climate change hazard (change in snow conditions), for which snowmaking is an adaptation measure affecting the hazard level. The study does not address the vulnerability component, nor the exposure (ski resorts remain at the same location). We have thus rephrased the text, where the term "risk"*

*appeared, for better clarity, and added a concluding paragraph which better reflects on all dimensions of the socio-ecological risks under climate change.*

As the other reviewer pointed out, the term "mitigation" should be strictly avoided in this context, since mitigation is commonly use to denote activities that aim to avoid or minimize climate change. Technical snowmaking can rather be called an adaptation to climate change.

*Similar to the response to Reviewer #1, we agree that the term "mitigation" needs to be replaced, in order to not create confusion with the use of the term in the climate change literature..*

---

## Author Comment (AC3) · 17 Mar 2019

**1   General comments**

In this paper results from snowpack modeling tailored to ski resort operations and potential impacts of climate change are presented.  From a regional perspective it is one of the view climate impact analyses for the ski industry in France, one of the most important ski markets in the world which has so far been under-researched. The applied model was already introduced elsewhere, but to my knowledge this is the first application for assessing the future perspectives of the ski industry. It is an important

contribution as snowmaking - an important adaptation strategy - was included. I have some question in the methods section (see comments below) and I suggest to include a paragraph in the discussion section on the "take-home message" of the paper. Apart from % changes illustrated in the results section, is it possible to evaluate the (near or longer term) future for skiing tourism in France and the Spanish Pyrénées? Is climate change a serious challenge, or manageable?

*We thank Robert Steiger for positive feedback and constructive suggestions. Please find below our replies to the specific points of the review comment. We have not included a "take-home message" section in the Discussion, but the Conclusion was refined to better convey the key results from this study, in terms of snow reliability. The revised conclusion highlights that snow reliability is not the only driver for ski resorts sustainability.*

**2 Specific comments**

Specific comments p. 3, l. 2: Damm et al. (2017) did not include snowmaking in their assessment, so the reference does not match your statement "...and the snowmaking requirements so as to compensate the loss over Europe (Damm...).

*We agree with this comment and have revised the sentence so that this error is corrected.*

p.4, l.5: what is the justification for using the village elevation?

*This statement in the manuscript refers to our method to compute the village elevation*

*in the ski resorts, based on the location of housing infrastructure. We indeed compute the elevation of the village, which corresponds to the locations where tourism housing infrastructures are located. Snow conditions in the immediate vicinity of housing infrastructure are critical for ski resort managers, because this is where tourists access the ski area, at the lower part of major ski lifts. A previous study (Spandre et al., 2016, J. Outdoor Tour. Recreat.) has demonstrated that ensuring appropriate snow conditions at the village elevation is a critical motivation for snow management in ski resorts, hence the focus on this elevation band.*

p. 4, l.7-15: It is not clear to me what data was available in which region and which data you had to estimate. I understand that all data (village elevation, min/max elev., ski lift power and surface area) was available for France. As seen in Fig 1 you then estimated ski lift power (?) based on the surface area you drew from OSM and on the linear model derived from French ski resorts? Then you also had to estimate the elevations of Spanish/Andorran ski areas? Why that if you had OSM data? How can you explain the outliers in the OSM/BD stations figure?

*We have clarified the wording of this part of the document to avoid ambiguities. Had the study only concerned French ski resorts, we would not have used OpenStreetMap (OSM) data. STRMTG data (detailed ski lifts characteristics) and high resolution digital elevation models and IGN data, used for the work on estimating the village elevation, were not available to use over Spain and Andorra. Based on OSM data for French ski resorts, we have analyzed the relationship between ski resorts surface area and ski resorts indicators such as ski lift power, and we have used this relationship to infer this information for ski resorts in Spain and Andorra, assuming that the general relationship established over a wide range of French ski resorts from the Alps and Pyrenees is most likely to hold in Spain and Andorra, where no major structural difference exists with French ski resorts. We have not analyzed specifically the outliers, because the*

*deviations to the main regression line where generally small compared to the full elevation range - this could be investigated in a future study.*

p. 5, l. 12: can you add some data on Spain as well? (the ski areas in this study represent xy% of ski lift infrastructures of Spain)

*We have added more information on the share of the Spanish and Andorran ski tourism infrastructure covered in our study.*

p. 6, l. 11: 150 kg/m2 -> if it is density it should be kg/m3 ; this is an uncommon density for technical snow for base layer snowmaking, typically it is around 400 kg/m3

*The values provided in the text are indeed snow water equivalent values (SWE), expressed in their native physical unit kg per unit surface area ($kg\,m^{-2}$). SWE is sometimes referred to using mm w.e., because liquid water density is 1000 $kg\,m^{-3}$. Using managed snow density of 500 $kg\,m^{-3}$, 150 $kg\,m^{-2}$ correspond to 30 cm snow depth. We have clarified this in the revised manuscript.*

p. 8, l. 21-23: please explain why the village elevation is relevant in your assessment. Later on you refer to the "lowest elevation of the ski area", this would be a clear explanation.

*As explained above, the village elevation corresponds to the mean elevation of tourism housing infrastructure. It does not necessarily correspond to the lowest elevation of the resort, because quite often, in the case of high elevation ski resorts, the lower elevation of the ski resorts is found below the tourism housing infrastructure.*

p. 9, l. 2. "snowmaking is limited to the lowest elevation and for a minority of seasons" -> I don't understand this sentence. How is snowmaking limited to a minority of seasons? Does that mean that snow is only produced in some years?

*We have clarified this statement, which now reads, in the revised manuscript : "Snowmaking is generally employed only at the lowest elevations, and it makes a difference only for a minority of seasons where natural snow conditions are too scarce." Indeed, here we emphasize on the fact that the 3 categories 1, 2 and 3 correspond to situations where snowmaking does not play a major role in determining the snow reliability of the ski resort.*

p. 11, l. 4: "to decrease in the Pyrenees, up to 15%" -> here the sentence structure confused me a bit because "decrease" is followed by a positive number and in the same sentence there is 15% another time, but as increase. Maybe consider to split this sentence in two?

*We agree and have reworded the sentence as follows: "The production of machine made snow at the snow reliability line is projected to remain steady or to decrease in the Pyrenees, up to 15% compared to the reference period. In the Alps, the production of machine made snow is projected to increase for all scenarios up to 15%."*

p. 11, l. 5: what do you mean by "either in the Northern or Southern Alps"?

*We have removed this part of the sentence, which was not necessary because it only meant to insists that similar changes were observed in the South and North of the*

*French Alps - which the meaning of the revised sentence captures, see above.*

Fig. 4/Discussion: how can you explain that the systematic bias is not existent in the Southern Alps?

*We have no specific explanation for this observation. That the distribution in ski resorts reliability categories is the same for past climate conditions using SAFRAN reanalysis and adjusted historical climate simulations is most likely a coincidence due to the combination of ski resorts elevations and snow simulations. Because this figure directly stems from counting ski resorts belonging to a given categories, it is most likely that the same distribution is found between the two cases, even though there are discrepancies between meteorological information of the SAFRAN and adjusted climate simulations. That there is a difference between the two can be viewed on Figure 3, which shows differences for all mountain regions, including the Southern Alps.*

---

## Editor Decision (ED1)

**Editorial comments**

Dear authors

Your paper is a very comprehensive assessment of the climate change impact on ski resorts using state-of-the-art methods.

I am very pleased with the revisions you made and recommend accepting your manuscript once you have addressed some minor technical corrections.

Page 5, Fig. 1: Please explain in the caption what "BD Stations" refers to. "Ski lift power" on the left y-axis.

Page 6, Fig. 2: Please use the same units for ski lift power as in the main text: km pers h$^{-1}$. In the caption: ski resort elevation

Page 16, Fig. 5: Please use the same units for ski lift power as in the main text: km pers h$^{-1}$.

*Some editorial suggestions:*

Page 1, line 8: While 99% of ski lift infrastructures are reliable for snow … may face difficulties in the near future.

Page 1, line 10: … with either steady conditions …

Page 1, line 17: This prompts the question of how climate change affects ski resorts and the ability of snow making as adaptation measure.

Page 1, line 19: "100-days rule"  (cf. page 7, line 14).

Page 2, line 6: … single point representations…

Page 2, line 12: elevations, between which 75% …

Page 3, line 25, ski lift installation

Page 4, line 3: the *mean ski lift elevation* … and several times elsewhere in the manuscript

Page 4, line 13: ski resort characteristics (same as *ski lift infrastructures*)

Page 5, line 14: $1.2 \times 10^{-3}$

Page 6, line 12: elevation

Page 7, lines 20-21: in cases when the season length … was longer, shorter …. Elevation

Page 9, line 17: season when natural snow conditions

Page 10, line 3: Due to snowmaking the median elevation increases by 700 m in …

Page 11, line 8: … suffer from the decrease in periods suitable for snow making.

Page 13, line 12: critical situation

Page 17, line 14-15: … might be considered in a critical situation since technical reliability cannot…

Jürg Schweizer

---

## Author Response (AR2)

CNRM UMR 3589

Samuel Morin
Météo-France – CNRS
Centre National de Recherches Météorologiques
Centre d'Etudes de la Neige
1441 rue de la piscine
F-38400 St Martin d'Hères

Object: Submission of revised manuscript "Winter tourism under climate change in the Pyrenees and the French Alps: relevance of snowmaking as a technical adaptation" to *The Cryosphere*.

Grenoble, April 3, 2019

Dear Editor,
Dear Jürg,

Please find enclosed our revised submission to *The Cryosphere* entitled "Winter tourism under climate change in the Pyrenees and the French Alps: relevance of snowmaking as a technical adaptation" by Pierre Spandre et al.

We have performed are requested changes in the text and figures. We take this opportunity to thank again the Editor for his comments and work in handling our manuscript.

Sincerely,

Samuel Morin, on behalf of the author team